# Opportunistic Expert Activation: Batch-Aware Expert Routing for Faster Decode Without Retraining

**Costin-Andrei Oncescu** [1 2]   **Qingyang Wu** [2]   **Wai Tong Chung** [2]   **Tsai-chuan Wu** [2]   **Bryan Gopal** [2]
**Junxiong Wang** [2]   **Tri Dao** [3 2]   **Ben Athiwaratkun** [2]

## Abstract

An increasing number of LLMs employ Mixture-of-Experts (MoE) architectures where the feedforward layer is replaced by a pool of experts and each token only activates a small subset of them. During autoregressive generation, these models often enter a memory-bound regime even for moderate batch sizes because the average expert load grows more slowly than in an equivalent dense feedforward layer. Consequently, MoE latency is governed by the number of activated experts. We introduce a framework for **dynamically** re-routing token-to-expert mapping to lower this number (and thus, the decode latency) while preserving a comparable quality. Our best results use a **batch-aware routing** that works by having tokens **piggyback** experts that have already been loaded into memory due to being crucial to other tokens within the same batch. At batch size 16, OEA reduces MoE-layer decode latency by 39% on Qwen3-30B while preserving standard-error-adjusted downstream accuracy, and by 15% on Qwen3-235B with only small overall degradation on the long-generation benchmark suite.

## 1. Introduction

Mixture-of-Experts (MoE) architectures have contributed significantly to the state-of-the-art in language modeling (Liu et al., 2024a; Kimi Team et al., 2025; Yang et al., 2025). They replace the feedforward layer with a pool of experts (smaller feedforward layers) and route each input to only a small subset of the pool. By employing this sparse and conditional computation, MoEs decouple model size from

computation cost, enabling more amenable model scaling.

When deploying these models, serving frameworks (Kwon et al., 2023; Zheng et al., 2024) usually batch several requests and proceed in two steps: prefill and decode. During the prefill stage, prompts are processed together in parallel across sequence length, much as a normal forward pass would. Then, decoding is the process of sequentially (autoregressively) generating one new token at a time, in parallel across a batch. This stage has a lower arithmetic intensity than prefill and is often memory-bound (Rajbhandari et al., 2022), where runtime is limited by data movement bandwidth rather than arithmetic throughput.

Because decoding dominates serving time for long sequences and interactive workloads, reducing its latency directly improves user experience and cost efficiency.

**The problem.** During decoding, it takes a larger batch size to get into a regime where experts are not memory-bound. This is because when each token activates $k$ experts out of $N$, the average per-expert load increases only at a rate of $k/N$ which is low by design in MoEs (e.g. $1/16$ in Qwen3). Coupled with the arithmetic intensity being roughly 100-200 (NVIDIA, 2022), the sparsity factor $N/k$ results in required batch sizes of order of thousands for MoEs to be in compute bound regime (e.g. $\approx 1.6k$ for Qwen3). Hence, for moderate batch sizes, the latency of an MoE layer is not dominated by the computational load of individual experts, but rather by the overhead of fetching the weights of all activated experts from the high-bandwidth memory (HBM) to the on-chip one (SRAM) (Rajbhandari et al., 2022). Consequently, latency becomes effectively linear in the number of unique activated experts, a number that can grow quickly with batch size in spite of each token activating only a few experts; this is because we need to activate the *union* of all these small sets of experts.

This paper introduces Opportunistic Expert Activation (OEA), a batch-aware routing framework designed to lower decode latency by explicitly minimizing the number of unique active experts per batch during inference. OEA operates without any model retraining and comprises two stages: i) OEA sets a minimum quality baseline for each

---

[1]Harvard University. Part of the work was done when Costin was interning at Together AI. [2]Together AI [3]Princeton University. Correspondence to: Costin-Andrei Oncescu <concescu@g.harvard.edu>.

*Proceedings of the $43^{rd}$ International Conference on Machine Learning*, Seoul, South Korea. PMLR 306, 2026. Copyright 2026 by the author(s).

token by keeping the first few of its expert choices to guarantee crucial computations take place, and ii) augments this baseline by routing tokens to additional, lower-priority experts only if those experts already need to be loaded due to another token's baseline requirement within the same batch.

This "piggybacking" mechanism allows the model to recover some of the performance that is potentially lost due to activating fewer experts, practically for free since it preserves the number of activated experts (and thus latency).

**Relation to Prior Work.** OEA is complementary to approaches that reduce the number of experts activated *per token* (Lu et al., 2024). In contrast, our piggybacking phase can be applied on top of such methods at no added cost. Unlike prior dynamic batch-aware routing strategies (Gupta et al., 2024), OEA guarantees a batch-independent quality baseline for every token, ensuring consistent per-token computation regardless of batch composition.

**Contributions** Our contributions are as follows:

1. We formalize the MoE decode latency problem under a memory-bound roofline model, showing that reducing the number of unique active experts is the primary optimization target.

2. We propose OEA, a dynamic routing algorithm that provides a tunable trade-off between model quality and system performance.

3. We evaluate OEA on the Qwen3-30B and Qwen3-235B models, demonstrating its ability to substantially reduce the number of active experts and, consequently, MoE latency, with minimal performance degradation on either downstream tasks or language modeling perplexity. At a batch size of 16, OEA achieves latency reductions of 39% on the 30B model and 15% on the 235B model.

## 2. Background and Motivation

Modern state-of-the art MoE models such as Kimi K2 (Kimi Team et al., 2025), Deepseek-V3 (Liu et al., 2024a) and Qwen3 (Yang et al., 2025) fundamentally incorporate the same setup (excluding potentially shared experts) popularized by Shazeer et al. (2017) — namely, they replace the feedforward layer of a transformer with sets of $N$ experts $E_1, \ldots E_N : \mathbb{R}^D \to \mathbb{R}^D$ (where $D$ is the embedding dimension) and a router scoring function $R : \mathbb{R}^D \to \Delta^N$ that assigns a normalized score $R(\boldsymbol{x})_i$ to each expert $i$. The output of the MoE module is then computed via:

$$\text{moe}(\boldsymbol{x}) = \sum_{i \in S} \frac{R(\boldsymbol{x})_i}{\sum_{j \in S} R(\boldsymbol{x})_j} E_i(\boldsymbol{x}) \quad (1)$$

where $S = \text{Top}_k(R(\boldsymbol{x}))$ is the set of indices of top-$k$ values of the router's scores. The extra normalization factor is optional, but enabled in Qwen3, the model we evaluate. Henceforth, we use $B$ for batch size.

Typically, serving these models is done by batching requests and proceeding in the following two stages: (i) the **prefill stage**, where activations and KV caches are computed for the prompts. This passes over entire prompts' at once, thus increasing the effective (token) batch size (*i.e.*, sequence length × batch size) of the MoE layers, resulting in heavier loads for each expert; and (ii) the (iterative) **decode stage** where, at a given decode step, exactly one token of each sequence in the batch is processed for next-token prediction. Crucially, the effective batch size passed to the MoE layers is now only equal to the batch size.

Note that the low effective (token) batch size seen during decoding is further exacerbated by the fact that the average expert load only increases at a rate of $k/N$ per token. This raises the threshold batch size for reaching the compute-bound regime, implying that even moderately-sized batches can still result in memory-bound experts. In this regime, the time to fetch expert's weights from HBM into on-chip SRAM dominates the time needed to compute $E_i$'s outputs. Consequently, for each expert, latency depends primarily on whether it is activated at all: if no token is routed to it, its weights need not be fetched and thus incur no latency, whereas once it is activated and fetched, the marginal cost to serving additional tokens is negligible. Therefore, when experts are not executed in parallel, overall MoE latency scales with the number of activated experts.

To illustrate how quickly this quantity grows, consider, for example, a setting where each token activates $k = 8$ experts out of $N = 128$ total ones (as is the case for Qwen3). For a batch size of 16 tokens, any number of experts between 8 and 128 could be employed. Assuming uniform routing (which the models are trained to balance), the expected number of activated experts is $82$[1]. Note that this represents an increase of up to $10\times$ over a batch size of 1 (where the token only triggers $k = 8$ experts). This is not the case in non-MoE architectures where both a batch size of 1 and one of 16 would enter a memory-bound regime and thus incur a fixed one-time fetching cost.

In summary, reducing the number of activated experts directly targets the dominant term of MoE decode latency in the memory-bound regime. We discuss related work in Section 5 and place our method in perspective.

---

[1]The exact formula is $N(1 - (1 - \frac{k}{N})^B)$ where $B$ is the batch size.

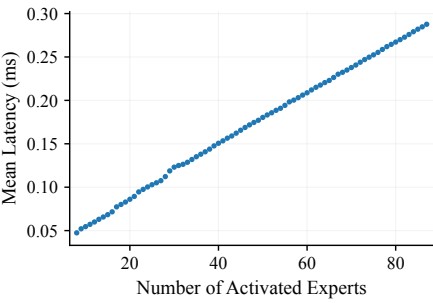

*Figure 1.* Mean MoE latency as a function of the number of activated experts within a decode batch. The average is computed over all layers and decode steps across a GPQA evaluation of the vanilla Qwen3-30B-A3B model.

## 3. Our framework

### 3.1. Latency and Number of Activated Experts

To formalize the argument introduced in Section 2, we adopt a simplified latency model for the computation performed by one expert. Let $f(n)$ represent the time it takes an expert to process $n$ tokens and let it be given by $f(0) = 0$ and $f(n) = an + b$ for $n > 0$. Here, $b$ is the cost of fetching the expert's weights from the high-bandwidth memory (HBM) into on-chip SRAM, while $a$ is the computation time it takes to process one token. It follows that the total latency of a whole MoE block is given by:

$$\sum_{i=1}^{N} f(\text{cnt}_i) = \sum_{i=1}^{N} b \cdot \mathbb{1}_{\text{cnt}_i > 0} + a \cdot \text{cnt}_i$$
$$= b \cdot T + a \cdot Bk \quad (2)$$

where $\text{cnt}_i$ is the number of tokens routed to expert $E_i$; $T$ is the number of experts that have at least one token routed to them; $B$ is the batch size; $N$ is the total number of experts; and $k$ is the number of experts activated per token.

Equation 2 shows that the overall latency is given by a memory-bound term linear in the number of active experts $T$ and a compute-bound term linear in the total computation load $Bk$. Whether we are in a compute- or memory-bound regime only indicates which of these terms dominates, but as a general statement, it directly follows that reducing $T$ lowers the latency. If the loads $\text{cnt}_i$ are small enough to be in a memory-bound regime (Rajbhandari et al., 2022), the total latency is dominated by $b \cdot T$ and thus we can expect almost proportional gains to the drop in $T$.

While this description is a simplification — it does not account for system-level effects such as kernel launch overhead, padding to equalize expert loads, or the use of optimized kernels like Grouped GEMM (Hejazi, 2024) — these factors do not alter the constraint. Grouped GEMM can improve efficiency by batching computations for different experts, but it still requires all activated expert weights to be

---

**Algorithm 1** OEA Routing Algorithm

1: **Input:** Token embeddings $\boldsymbol{x}_{1..B}$, Router scores $R(\boldsymbol{x}_i)$, Sorted expert indices $e_{i,j}$ for each token $i$ and rank $j$. Hyperparameters: $k_0, p, k^{\text{max}}, \text{maxP}$.
2: {Phase 1: Determine Baseline Experts}
3: **for** $i = 1$ **to** $B$ **do**
4:    Find $t_i = \min\{t' \mid \sum_{j=1}^{t'} R(\boldsymbol{x}_i)_{e_{i,j}} \geq p\}$
5:    $n_i \leftarrow \min(k_0, t_i)$ {Number of baseline experts}
6:    $S_i^{\text{base}} \leftarrow \{e_{i,1}, \ldots, e_{i,n_i}\}$
7: **end for**
8:
9: {Phase 2: Opportunistic Piggybacking}
10: $S^{\text{base}} \leftarrow \bigcup_{i=1}^{B} S_i^{\text{base}}$ {Union of all required experts}
11: **for** $i = 1$ **to** $B$ **do**
12:    $S_i \leftarrow S_i^{\text{base}}$ {Initialize final set with baseline}
13:    **for** $j = n_i + 1$ **to** maxP **do**
14:      **if** $|S_i| \geq k^{\text{max}}$ **then**
15:       **break**
16:      **end if**
17:      **if** $e_{i,j} \in S^{\text{base}}$ **then**
18:       $S_i \leftarrow S_i \cup \{e_{i,j}\}$
19:      **end if**
20:    **end for**
21: **end for**
22: **Output:** Final expert sets $S_1, \ldots, S_B$

---

loaded into on-chip memory, meaning the latency remains fundamentally tied to $T$ in the memory-bound regime. We confirm this empirically (Figure 1).

Finally, note that $\text{cnt}_i \leq B$, since each token can route to an expert at most once; this holds for any potential re-routing as well. Furthermore, for the original top-$k$ routing, if we are to further assume it to be uniform, it follows that $\mathbb{E}[\text{cnt}_i] = Bk/N$ which is much lower and thus increases the threshold for $B$ to be in a compute-bound regime. We henceforth turn our focus on optimizing $T$ and assume we are in a regime where this translates (as shown empirically) to lower overall latency.

To achieve this, we modify token routing during inference while preserving empirical performance. This approach is motivated by recent studies demonstrating the robustness of MoE models to re-routing (Li et al., 2025; Gupta et al., 2024). While several approaches have explored *static expert pruning* (Lu et al., 2024; Liu et al., 2024b) — permanently removing experts to save memory — this inevitably constrains the model's capacity. In contrast, our goal is to develop methods that maintain a minimum level of performance in the worst case while enabling full recovery of the model's original performance in the best case.

### 3.2. The Proposed Routing Algorithm

**Algorithm 2** *Simplified* OEA Routing Algorithm

---

1: **Input:** Token embeddings $\boldsymbol{x}_{1..B}$, Initial number of experts per token $k$, Sorted expert indices $e_{i,j}$ for each token $i$ and rank $j$. Hyperparameter: $k_0$.
2: {Phase 1: Determine Baseline Experts}
3: **for** $i = 1$ **to** $B$ **do**
4: $\quad S_i^{\text{base}} \leftarrow \{e_{i,1}, \ldots, e_{i,k_0}\}$
5: **end for**
6:
7: {Phase 2: Opportunistic Piggybacking}
8: $S^{\text{base}} \leftarrow \bigcup_{i=1}^{B} S_i^{\text{base}}$ {Union of all required experts}
9: **for** $i = 1$ **to** $B$ **do**
10: $\quad S_i \leftarrow S_i^{\text{base}}$ {Initialize final set with baseline}
11: $\quad$ **for** $j = k_0 + 1$ **to** $N$ **do**
12: $\quad\quad$ **if** $|S_i| \geq k$ **then**
13: $\quad\quad\quad$ **break**
14: $\quad\quad$ **end if**
15: $\quad\quad$ **if** $e_{i,j} \in S^{\text{base}}$ **then**
16: $\quad\quad\quad S_i \leftarrow S_i \cup \{e_{i,j}\}$
17: $\quad\quad$ **end if**
18: $\quad$ **end for**
19: **end for**
20: **Output:** Final expert sets $S_1, \ldots, S_B$

---

**Why two algorithms?** While Algorithm 1 describes our method — OEA — in its *full generality*, following exhaustive experiments, we conclude that a *simplified* version of it, described in Algorithm 2 (differences in red), recovers its performance while requiring fewer hyperparameters. We hereby describe its most general form and then touch on how to simplify at the end of Section 4.1.

Following Section 3.1, OEA aims to minimize the number of activated experts $T$ within a decode batch. Its core constraint is to ensure that the overall response quality, for any given sequence in the batch, does not significantly degrade. This motivates a two-stage approach that works by first establishing a *batch-independent* per-token compute floor, and then opportunistically recovering lost performance by exploiting the shared computation within the batch.

**Notation.** Suppose the $B$ tokens in the batch are $\boldsymbol{x}_1 \ldots \boldsymbol{x}_B$, and that their sorted expert index scores are $e_{i,j}$ where $e_i$ is a permutation such that,

$$R(\boldsymbol{x}_i)_{e_{i,1}} \geq R(\boldsymbol{x}_i)_{e_{i,2}} \geq \cdots \geq R(\boldsymbol{x}_i)_{e_{i,N}},$$

where $e_{i,j}$ represents the $j^{\text{th}}$ expert choice of $i^{\text{th}}$ token. In particular, the default router (as described in Section 2) routes token $\boldsymbol{x}_i$ to experts in the set $\text{Top}_k(R(\boldsymbol{x}_i)) = \{e_{i,1}, \ldots, e_{i,k}\}$. Our target is to decide sets $S_1, \ldots, S_B \subseteq \{1, \ldots N\}$ where $S_i$ represents the set of experts that the $i^{\text{th}}$ token routes to.

**Phase 1: Baseline expert selection.** The first phase guaran-

tees a minimum foundation for each token, irrespective of how it is batched. For each token $\boldsymbol{x}_i$, we create this baseline by activating the first $n_i$ experts, thus creating a base set of experts $S_i^{\text{base}} = \{e_{i,1}, \ldots, e_{i,n_i}\}$. This is motivated by empirical findings that the top-ranked experts are disproportionately critical to output quality (Gupta et al., 2024). The number of base experts is determined by two hyperparameters: (1) a fixed upper bound $k_0 \in \{1 \ldots N\}$; and (2) a cumulative score $p \in (0, 1]$, following $n_i = \min(k_0, t_i)$ where $t_i$ is the minimum number of experts it takes to reach a cumulative mass of $p$, such that,

$$\sum_{j=1}^{t_i - 1} R(\mathbf{x}_i)_j < p \leq \sum_{j=1}^{t_i} R(\mathbf{x}_i)_j.$$

Intuitively, $t_i$ is a function of the normalized scores — it is defined exactly as in Huang et al. (2024). While their work pretrained a model with a regularizer factor to ensure that $n_i$ is low on average, no such guarantee is assumed here. And thus, we decide to further cap $n_i$ by $k_0$. In general, it should never help to set $k_0 > k$ where $k$ is the model's default configuration. Finally, note that by setting $p = 1$, we essentially have a fixed $k_0$ and by setting $k_0 = N$, we practically have the top-$p$ method of Huang et al. (2024), so we generalize and abstract on both methods. We decided to adopt this approach to allow the number of experts to be adaptive to the router scores so that harder instances can demand more experts.

This $(k_0, p)$-heuristic can select experts that are critical to at least one token's predictions, and therefore the set of all essential experts $S^{\text{base}} = \cup_{i=1}^{B} S_i^{\text{base}}$ to activate.

**Phase 2: Opportunistic piggybacking.** Instead of adding any new experts into the mix, this second phase opportunistically recovers some performance by allowing tokens to *piggyback* onto experts already included in $S^{\text{base}}$, thus maintaining the number of activated experts $T = |S^{\text{base}}|$. For each token $i$, we traverse experts in decreasing order of their scores and select those in $S^{\text{base}}$ provided that (1) the number of selected experts does not exceed $k^{\text{max}}$ and (2) the expert's rank does not fall below a threshold position maxP. These constraints ensure that the selected experts do not degrade performance, either by over-diversifying expert usage or by selecting experts poorly aligned with the current token.

**Weighting after rerouting.** Once the routing sets $S_i$ are chosen, we keep the model's original router scores and renormalize them following Equation (1). Intuitively, this preserves the model's learned preferences among the experts we keep, while ensuring mixture weights still sum to 1. Other choices like using the weights of top-$k$ are possible, but we leave such optimization to future work.

# 4. Experimental Setup And Empirical Results

## 4.1. Cross-Entropy Experiments

**Motivation.** We use cross-entropy on a pretraining dataset as a granular proxy for the compound effect of our router intervention. We do this for two reasons. First is that it is much cheaper to measure cross-entropy than downstream performance thanks to its parallel computation. Thus, we can perform a large hyperparameter sweep to determine the *optimal* setting of OEA. Based on these findings, we suggest a simplified version of OEA (Algorithm 2) that we then evaluate on standard benchmarks in Section 4.2. Secondly, Unlike benchmarks, cross-entropy provides a more statistically reliable estimate of modeling quality since it provides dense, per-token signal rather than sparse, task-level evaluation.

**Dataset.** In the first round of experiments, we evaluated cross-entropy loss on a subset of the FineWeb-Edu dataset (Penedo et al., 2024), which we selected as a high-quality and diverse proxy since Qwen3's pretraining data is not public. We randomly selected 2048 sequences, each containing at least 8192 tokens to ensure a fixed batch size across positions since OEA is sensitive by construction to batch size. In particular, a batch size of 1 makes the piggy-backing redundant.

**Methodology.** We simulate $L$ decoding steps (sequence length) but execute them efficiently in parallel. At step $t$, we form a batch from the $t$-th token of each sequence and run routing *only within that step*: both the Phase 1 pruning and Phase 2 piggybacking are computed using tokens that share the same position $t$. No information (experts or scores) is shared across different positions, so piggybacking never crosses decode steps. We then process all steps in parallel by grouping expert workloads post-routing, which yields the same routing decisions as true sequential decode while enabling a fast, batched implementation for measurement. Throughout this computation, we track the *average number of activated experts* across positions and layers, as well as *average cross-entropy*.

**Experiments.** The parallel speedup allows for a comprehensive sweep of hyperparameters: $k_0 \in \{4, 5, 6, 7, 8\}$, $k^{\max} \in \{7, 8, 9, 10, 11\}$, $p \in \{0.4, 0.5, 0.6, 0.7, 0.8, 0.9, 1\}$ and maxP $\in \{8, 16, 32, 128\}$. On top of these, we also considered stopping after Phase 1 (for the same ranges of $k_0, p$) and forgoing the piggybacking — we refer to this as *Phase 1* or *pruned* routing. For each such routing algorithm, we swept $B \in \{8, 16, 32, 64\}$. We used 128 sequences and the full length of 8192 for $B = 8$ and cut sequence length by half for every batch doubling to keep the activation memory fixed and able to fit in the memory of a GPU. We also doubled the number of sequences for every batch doubling to keep the overall number of tokens fixed at 1M.

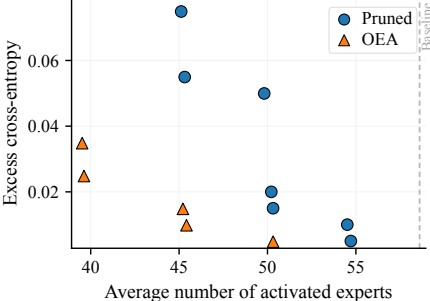

*(a)* Pruned vs OEA.

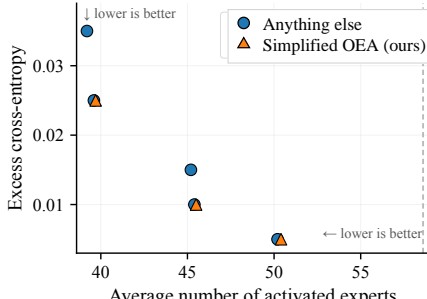

*(b)* Simplified OEA vs other configurations.

*Figure 2.* Cross-entropy delta relative to the baseline (lower left is better) at batch size $B = 16$. Points correspond to Pareto frontiers of different routing strategies. (a) OEA consistently outperforms pruned routing. (b) Simplified OEA achieves performance comparable to the best hyperparameter choices.

Equipped with these runs, we can investigate the effects of our design choices. There are four degrees of freedom corresponding to the four hyperparameters: $k_0$ and $p$ determine the pruning extent while $k^{\max}$ and maxP control whether adding an extra expert starts hurting.[2]

**Ablations.** For each experiment, we define its performance as a trade-off between cross-entropy and the average number of activated experts — the objective is to minimize both. To assess the effect of each hyperparameter value on performance, we plot the Pareto frontier of all experiments conducted with that value. As our purpose is to limit cross-entropy degradation, we plot the increase in cross-entropy with respect to a vanilla MoE and track across runs.

**Piggybacking gains.** Since our algorithm's core addition over a form of adaptive pruning is the piggybacking phase, the salient question is whether Phase 2 truly adds value: we answer this in the affirmative, as shown in Figure 2a.

We find three consistent patterns regarding hyperparameter choice. The ablation plots corresponding to them are available in the Appendix A.

---

[2]Note that setting $p = 1$ is equivalent to not using it at all and so is the case for maxP $= 128$.

1. **Using $p < 1$ does not help.** Setting $p = 1$ (equivalent to using top-$k_0$ in Phase 1) performs on par with $p < 1$ (Figure 8). This holds across both OEA and the partial "pruned" (Phase 1 only) case. We considered employing a top-$p$ scheme to allow the choice of base quality to be a function of the router scores, but there is no significant marginal gain from this adaptivity.

2. **$k^{\mathrm{max}} = k$ works best.** Interestingly, we find that bounding the number of experts per token at exactly $k$ works better than both smaller and larger choices (Figure 6). Naturally, one expects more experts to help but interestingly, using $k^{\mathrm{max}} = 9$ experts does not really improve above $k^{\mathrm{max}} = 8$; in fact, further increasing to $k^{\mathrm{max}} = 10, 11$ actually results in degradation.

3. **Setting $maxP < N$ does not help.** Our ablation over maxP (Figure 5) shows that refraining from piggybacking onto an activated expert due to it being too far down a token's preference list is detrimental for the optimal values of $k^{\mathrm{max}}$. It is worth noting that maxP could only make a difference when we do not have $k^{\mathrm{max}}$ activated experts in the top-maxP preferences of a token, which becomes less likely with the increase in $B$ (and thus $|S^{\mathrm{base}}|$). Finally, one important consequence of maxP $= 8$ strictly hurting is proving that using out-of-policy experts confers a strict advantage. This is contrary to the thesis that those experts are not trained to be useful for this one token.

**Simplifying OEA** Putting these together, we conclude that we can drop the usage of top-$p$ in Phase 1 and that of maxP in Phase 2, as well as set $k^{\mathrm{max}}$ to be $k$. This leaves us with a simplified version of the OEA routing presented in Algorithm 2. Figure 2b shows that our hyperparameter findings are jointly consistent, suggesting the simplified algorithm to be as performant as its general counterpart. A major benefit is thus the reduced cost of hyperparameter sweeps prior to deploying the model: $k_0$ controls both the guaranteed baseline quality and the drop in activated experts.

### 4.2. Downstream Evaluations

**Models** We benchmark our approach on both the Qwen3-30B and Qwen3-235B-A22B models. The latter doubles the number of layers (96), embedding dimension (4096) and expert hidden dimension (1536) while the same attention head configuration top-8/128 routing. All experiments are performed under tensor parallelism across 8 H100 GPUs within a single HGX H100 node interconnected via NVSwitch (18 NVLink per GPU pair).

**Setup.** We conducted downstream evaluation on four benchmarks: AIME24, MATH_500 (Hendrycks et al., 2021), GPQA (Rein et al., 2024) and LIVECODEBENCH V5 (Jain et al., 2025). All accuracy reported is an average over four

runs of each for Qwen3-30B (and three runs for Qwen3-235B), with the exception of AIME24 which we evaluate four times more runs (since it only has 30 data points and, thus, higher variance). For all runs, we use a temperature of 0.6, nucleus sampling with $p_{\mathrm{samp}} = 0.95$ and generate up to 32768 tokens. We integrate our router into the SGLang framework (Zheng et al., 2024). For each run, we track the batch size, number of activated experts and the latency for every layer and decode step. Note that, throughout the serving process, batch size can and does vary as requests are finished, enqueued or retracted. Since our routing algorithm only benefits latency in moderate batch size regimes, we use it only during decode, not prefill.

**Experiments.** Informed by our findings from Section 4.1, we only used the recommended settings and thus only tried the simplified algorithm (2) parameterized only by $k_0$. We tested all values of $k_0 \in \{3, 4, 5, 6, 7\}$ on Qwen3-30B, and all except $k_0 = 7$ on Qwen3-235B due to computational constraints. We also evaluated post-Phase 1 (pruned) routing (as in Section 4.1) for the same set of $k_0$ values. As batch size cannot be fixed in SGLang, we use its `--max-running-requests` option to set a maximum batch size. We could not do this for a batch size of 32 due to the large KV cache size, so we report OEA numbers for a batch size bounded at 16.

**MMLU-Redux** For completeness, Appendix A.1 reports additional MMLU-REDUX (Gema et al., 2025) results without thinking mode. We keep these results separate from the main downstream suite because their generations are substantially shorter and therefore probe a different serving regime; nevertheless, they provide useful evidence on general-knowledge accuracy and on the batch-size dependence of OEA.

**Piggybacking gains.** Tables 1 (a) and (b) report the average performances of the pruned (Phase 1) approach, of the simplified OEA, and of the base model ($k_0$=8) for the 30B and 235B model, respectively. We find that:

- **Qwen3-30B**: Across all benchmarks, except GPQA, using top-5 rather than top-8 experts does not make a statistically significant difference[3]. However, further lowering it to 4 and 3 starts showing substantial degradation. OEA manages to recover lost performance even for $k_0 = 3$ while fundamentally not incurring any extra cost over its pruned counterpart; this is the marginal gain of piggybacking and our main contribution.

- **Qwen3-235B**: In the basic pruning setup, performance drops sharply at $k_0 = 5$ falling below the base model by 15% on LIVECODEBENCH, 8% on GPQA. In con-

---

[3] A result $\mu \pm$ se is considered standard-error adjusted worse than $\mu_{\mathrm{vanilla}} \pm \mathrm{se}_{\mathrm{vanilla}}$ when $\mu + \mathrm{se} < \mu_{\mathrm{vanilla}} - \mathrm{se}_{\mathrm{vanilla}}$

*Table 1.* Ablation across $k_0$: Benchmark accuracies for Phase 1 (pruned, top-$k_0$) and simplified OEA routing (top-$k_0$+piggybacking) on (a) and (b). **Pruned** refers to using top $k_0$ experts per token, **OEA** does additional piggybacking and vanilla represents the default model. Results averaged over 3 runs. Setups that are no worse than vanilla (standard-error adjusted) are in bold.

*(a)* Qwen3-30B-A3B

| | $k_0 = 3$ | | $k_0 = 4$ | | $k_0 = 5$ | | $k_0 = 6$ | | $k_0 = 7$ | | VANILLA |
|---|---|---|---|---|---|---|---|---|---|---|---|
| BENCHMARK | PRUNED | OEA | PRUNED | OEA | PRUNED | OEA | PRUNED | OEA | PRUNED | OEA | |
| AIME24 | 51.2 | **80.0** | 75.8 | **81.9** | 80.6 | **81.5** | 80.2 | **80.8** | 82.5 | 78.5 | **80.4** |
| GPQA | 45.7 | **58.6** | 54.3 | **59.3** | 56.2 | **61.1** | 58.3 | **62.2** | 59.7 | **60.6** | **60.2** |
| LIVECODEBENCH | 37.4 | **61.2** | 58.2 | **62.7** | 63.2 | 62.0 | 63.1 | **63.1** | 63.0 | **62.5** | **62.1** |
| MATH_500 | 91.1 | **93.5** | 92.7 | **93.1** | 92.6 | **93.3** | 93.1 | **93.1** | 93.3 | **93.2** | **92.8** |

*(b)* Qwen3-235B-A22B

| | $k_0 = 3$ | | $k_0 = 4$ | | $k_0 = 5$ | | $k_0 = 6$ | | VANILLA |
|---|---|---|---|---|---|---|---|---|---|
| BENCHMARK | PRUNED | OEA | PRUNED | OEA | PRUNED | OEA | PRUNED | OEA | |
| AIME24 | 17.5 | 81.4 | 69.4 | 82.5 | 81.9 | **83.6** | 82.8 | **83.6** | **85.0** |
| GPQA | 43.8 | 66.3 | 56.4 | **67.7** | 60.6 | **67.5** | 64.1 | 67.5 | **68.4** |
| LIVECODEBENCH | 5.7 | 63.4 | 27.4 | 67.1 | 53.5 | 66.1 | 60.8 | 66.1 | **68.5** |
| MATH_500 | 80.9 | **94.4** | 93.3 | **94.8** | 94.5 | **94.7** | 94.5 | 94.3 | **94.7** |

trast, OEA at $k_0 = 5$ maintains performance on all benchmarks except for LIVECODEBENCH where its accuracy declines slightly by 2%.

**Relationship between latency and the number of activated experts.** The central hypothesis of this work (introduced in Section 3.1) is that for moderate batch sizes, the MoE latency scales linearly with the number of activated experts. To confirm this, we tracked all the ($T$, latency) pairs obtained at all decode steps and all layers. Figure 1 shows the average latency for a fixed number of activated experts across the whole GPQA run of the vanilla Qwen3-30B model (across decode steps and layers); the standard errors are all less than $2 \cdot 10^{-4}$ indicating that latency is well predicted by these means. Since the MoE module itself was left unchanged, this trend is independent of the routing strategy. Strikingly, the linear trend fits regression at $R^2 > 0.99$, thus affirming the thesis of this work: latency is linearly controlled by the number of activated experts.

**Latency gains via reducing active experts.** Building on the above, we examine how OEA's reduced expert activation translates to practical latency reductions. For the Qwen3-30B model, Table 2 reports the average number of active experts (aggregated over layers and decode steps) as a function of $k_0$, with $k_0 = 3$ halving the number of activated experts. This corresponds to latency reductions of 39% for $k_0 = 3$ and 23% for $k_0 = 5$; complete data is displayed in the Appendix (Table 9). For Qwen3-235B, the results in Table 3 show a 15% speedup at $k_0 = 5$; we attribute this smaller relative reduction to the additional overhead of tensor parallel's all-reduce.

## 5. Related Work

The challenge of optimizing MoE inference latency is an active area of research (Liu et al., 2024c). The modern MoE paradigm in Transformers was established by Shazeer et al. with the Sparsely-Gated MoE layer (2017), which employs a trainable gating network to route each token to a top-$k$ subset of experts. Their goal was to decouple the model parameters from the computational cost of training to enable scaling of larger LLMs. However, this approach introduces significant systems-level challenges. The most notable is **load imbalance** (where the router ends up favoring a subset of "popular" experts) leads to router collapse, leaving other experts and their associated parameters and hardware underutilized. To address this matter, load-balancing losses are employed during training (Shazeer et al., 2017; Fedus et al., 2022; Liu et al., 2024a).

More central to our work are the issues that arise in batched inference for large sparsity. Serving systems (Kwon et al., 2023; Zheng et al., 2024) rely on batching to achieve high throughput, but this forces the activation of the union of all experts selected by any token in the batch, quickly negating MoE's sparsity and deeming MoE layers to be memory-bound for moderate batch sizes. To address these fundamental issues, paradigms going beyond token-centric top-$k$ routing have been explored.

**Expert choice routing.** Zhou et al. (2022) inverted the selection logic, allowing each expert to select its top-$k$ preferred tokens from the batch. This is an inherently batch-aware mechanism that guarantees perfect load balancing by design, eliminating the need for auxiliary losses. While it enables a variable number of experts per token, its purpose is optimizing throughput via load balancing rather than minimizing

*Table 2.* Average number of activated experts when using **simplified OEA** (top-$k_0$ + piggybacking) on Qwen3-30B-A3B.

|  | $k_0 = 3$ | $k_0 = 4$ | $k_0 = 5$ | $k_0 = 6$ | $k_0 = 7$ | VANILLA |
|---|---|---|---|---|---|---|
| AIME24 | 22.2 | 26.5 | 30.5 | 36.0 | 39.2 | 43.0 |
| GPQA | 26.5 | 31.4 | 37.5 | 42.9 | 47.6 | 51.6 |
| LIVECODEBENCH | 23.8 | 28.7 | 33.9 | 38.7 | 42.5 | 47.2 |
| MATH_500 | 27.9 | 33.0 | 38.6 | 43.7 | 48.3 | 53.5 |
| AVERAGE | 25.1 | 29.9 | 35.1 | 40.3 | 44.4 | 48.8 |
| NORMALIZED AVERAGE | 0.51 | 0.61 | 0.72 | 0.83 | 0.91 | 1.00 |

*Table 3.* Average MoE layer latency (microseconds), with *all-reduce* for **simplified OEA** (top-$k_0$ + piggybacking) on Qwen3-235B-A22B.

|  | $k_0 = 3$ | $k_0 = 4$ | $k_0 = 5$ | $k_0 = 6$ | VANILLA |
|---|---|---|---|---|---|
| AIME24 | 86.4 | 92.6 | 98.8 | 105.7 | 118.4 |
| GPQA | 86.7 | 93.8 | 99.5 | 104.7 | 116.0 |
| LIVECODEBENCH | 88.2 | 95.3 | 102.8 | 108.6 | 121.1 |
| MATH_500 | 89.6 | 97.4 | 104.4 | 108.7 | 122.2 |
| AVERAGE | 87.7 | 94.8 | 101.4 | 106.9 | 119.4 |
| NORMALIZED AVERAGE | 0.73 | 0.79 | 0.85 | 0.90 | 1.00 |

the number of active experts to reduce latency.

**Architectural solutions (shared experts).** Models like DeepSeek-V3 (Liu et al., 2024a) and Kimi K2 (Kimi Team et al., 2025) incorporate a hybrid architecture with both "routed" and "shared" experts. The shared experts process every token in the batch, providing a form of guaranteed computational reused core. This allows for system co-optimization, such as hiding communication latency behind the shared expert's computation. Such approaches represent a static design solution to shared computation, contrasting with our dynamic, runtime approach that requires no architectural modifications.

OEA best fits under the paradigm of dynamic, inference-time optimizations that modify MoE behavior without retraining. Such approaches are crucially different from static pruning methods that permanently remove experts they expect to not be crucial to performance. While effective for compression, the behavior of the dropped experts cannot be recovered if a token depended on them. Such issues could be mitigated by making exclusion decisions adaptive to the batch, although then the model size cannot be reduced.

**Token-centric dynamic skipping.** One category of dynamic methods operates on a per-token basis. Lu et al. (2024) proposed dynamically skipping a secondary expert if its router score is significantly lower than the primary one, saving computation on a per-token basis. Similarly, the Online Dynamic Pruning (ODP) technique identifies less important tokens and assigns them fewer experts (Huang et al., 2025). These methods are not explicitly batch-aware and thus miss opportunities for shared computation.

**Expert offloading and prefetching.** In memory-constrained environments, model weights are stored on CPU and offloaded on demand. Systems such as Pre-gated MoE (Hwang et al., 2024) employ predictive prefetching, where information from the current layer is used to anticipate and pre-load experts for the next layer while the current layer's computation is underway. Read-ME (Cai et al., 2024) extends this line of work by using a shared pre-gating router across layers, enabling aggressive expert offloading and prefetching. eMoE (Tairin et al., 2025) is also related: it predicts and loads only the experts expected to be needed, exploiting recurring expert-routing patterns and task information to reduce GPU memory usage and loading latency. These systems optimize expert storage, offloading, and prefetching, whereas our method optimizes computational reuse within a single decode batch.

**Concurrent batch-aware active-expert reduction.** Several very recent works target the same memory-bound MoE decoding bottleneck. SERE (Wu et al., 2026) is particularly close to our setting: it also preserves critical experts for each token and then reduces the batch-level active expert set by rerouting secondary expert choices. The main algorithmic difference is that SERE uses an expert-similarity-based reconstruction rule, whereas OEA uses a simpler opportunistic heuristic: after fixing the top-$k_0$ per-token floor, each token piggybacks on already-active experts in router-score order until reaching the per-token cap $k^{\max}$. METRO (Yu et al., 2025) is also related, but focuses on expert-parallel serving: it argues that in the memory-bound regime, systems should balance activated experts rather than token counts across devices. This is complementary to our focus on modifying the within-batch routing decision itself. Overall, these concurrent works are closely aligned with our motivation and

highlight the same broader lesson: in memory-bound MoE decoding, the number and placement of activated experts can be more important than the nominal number of routed tokens.

Lynx (Gupta et al., 2024) is also closely related: it uses dynamic batch-aware expert selection to reduce the active expert set during MoE inference. Unlike OEA, Lynx primarily exposes a batch-level active-expert or latency-oriented knob, whereas OEA exposes a per-token baseline knob $k_0$. In particular, Lynx prunes experts from the batch-level union, so the resulting computation and quality are inherently batch-dependent; an expert may be removed even if it is important for a single token. In contrast, OEA is designed around a batch-independent per-token quality floor: each token always executes a token-specific baseline set of high-scoring experts, and batch information is used only in an additive piggybacking step that routes tokens to already-active experts. Because these knobs differ, a strict apples-to-apples comparison requires choosing an arbitrary mapping between a latency budget and a per-token quality floor. We therefore compare against the conservative baseline implied by our standalone-pruning phase (i.e. Phase 1 / top-$k_0$ routing) and isolate the additional quality recovery provided by piggybacking.

## 6. Conclusion

In this work, we introduce OEA, a new expert routing algorithm targeting the problem of memory-bound MoE decoding under moderate batch sizes. OEA achieves this by reducing the number of activated experts per decode batch. To do this, it first activates a few top experts per token deemed crucial to its performance and then opportunistically piggybacks some more that were crucial to other tokens in the batch. This approach results in MoE latency speed-ups of 39% and 15% for the Qwen3-30B and Qwen3-235B models, respectively, with only minimal benchmark accuracy degradation on the 235B model.

## Impact Statement

This paper presents work whose goal is to advance the field of Machine Learning. There are many potential societal consequences of our work, none which we feel must be specifically highlighted here.

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

*Table 4.* Ablation across $k_0$: Benchmark accuracies, *standard error included*, of **simplified OEA** routing (top-$k_0$+piggybacking) on Qwen3-30B-A3B. Vanilla represents the default model.

|  | $k_0 = 3$ | $k_0 = 4$ | $k_0 = 5$ | $k_0 = 6$ | $k_0 = 7$ | VANILLA |
|---|---|---|---|---|---|---|
| AIME24 | $80.0 \pm 0.59$ | $81.9 \pm 0.71$ | $81.5 \pm 0.62$ | $80.8 \pm 0.90$ | $78.5 \pm 1.33$ | $80.4 \pm 0.99$ |
| GPQA | $58.6 \pm 1.22$ | $59.3 \pm 0.15$ | $61.1 \pm 1.69$ | $62.2 \pm 0.56$ | $60.6 \pm 0.36$ | $60.2 \pm 0.83$ |
| LIVECODEBENCH | $61.2 \pm 0.94$ | $62.7 \pm 0.53$ | $62.0 \pm 0.33$ | $63.1 \pm 0.57$ | $62.5 \pm 0.47$ | $62.1 \pm 0.94$ |
| MATH_500 | $93.5 \pm 0.29$ | $93.1 \pm 0.29$ | $93.3 \pm 0.27$ | $93.1 \pm 0.30$ | $93.2 \pm 0.08$ | $92.8 \pm 0.22$ |

*Table 5.* Ablation across $k_0$: Benchmark accuracies, *standard error included*, of **pruned** routing (top-$k_0$) on Qwen3-30B-A3B. Vanilla represents the default model.

|  | $k_0 = 3$ | $k_0 = 4$ | $k_0 = 5$ | $k_0 = 6$ | $k_0 = 7$ | VANILLA |
|---|---|---|---|---|---|---|
| AIME24 | $51.2 \pm 1.42$ | $75.8 \pm 0.83$ | $80.6 \pm 0.86$ | $80.2 \pm 0.40$ | $82.5 \pm 0.83$ | $80.4 \pm 0.99$ |
| GPQA | $45.7 \pm 0.84$ | $54.3 \pm 0.53$ | $56.2 \pm 0.43$ | $58.3 \pm 0.25$ | $59.7 \pm 2.01$ | $60.2 \pm 0.83$ |
| LIVECODEBENCH | $37.4 \pm 0.83$ | $58.2 \pm 1.41$ | $63.2 \pm 0.95$ | $63.1 \pm 0.76$ | $63.0 \pm 0.77$ | $62.1 \pm 0.94$ |
| MATH_500 | $91.1 \pm 0.37$ | $92.7 \pm 0.26$ | $92.6 \pm 0.13$ | $93.1 \pm 0.29$ | $93.3 \pm 0.26$ | $92.8 \pm 0.22$ |

# A. More benchmark results

Tables 4 and 6 show the benchmark accuracies of *simplified OEA* together *with the standard errors* across the 4 Qwen3-30B, respectively 3 Qwen3-235B runs.

Tables 5 and 7 show the benchmark accuracies for the *pruned* routers (top-$k_0$) *with the standard errors* across the 4 Qwen3-30B, respectively, 3 Qwen3-235B runs.

Furthermore, Table 8 shows the average number of activated experts while using simplified OEA routing on Qwen3-235B.

For the Qwen3-235B model, we report, in Figure 3, the average latency (across decode steps and layers) for a fixed number of activated experts across a complete run of GPQA.

## A.1. MMLU-Redux without thinking mode

To complement the long-generation downstream benchmarks in Section 4.2, we additionally evaluate MMLU-Redux without thinking mode. We report these results in the appendix because the generations are substantially shorter than those of AIME24, GPQA, MATH 500, and LiveCodeBench under thinking mode, and therefore the setting is not directly comparable to the main long-CoT evaluation. The shorter generations, however, make MMLU-Redux useful for testing general-knowledge accuracy and for studying larger batch sizes without the same KV-cache pressure.

Tables 10 and 11 show that OEA follows the same qualitative pattern as in the main evaluation. On Qwen3-30B-A3B, OEA remains close to vanilla across a wide range of $k_0$ values while reducing normalized MoE-layer latency. On Qwen3-235B-A22B, OEA at $k_0 = 6$ is not standard-error-adjusted worse than vanilla, while smaller $k_0$ values expose a stronger latency–quality tradeoff.

We also use MMLU-Redux to isolate the effect of batch size on Qwen3-235B-A22B. This experiment is feasible because the non-thinking-mode generations are much shorter than in the main long-CoT suite. As shown in Table 12, OEA generally recovers more of the pruning-induced accuracy loss at larger batch sizes, because the Phase-1 expert union grows and creates more piggybacking opportunities. At the same time, the relative latency gain decreases as batch size increases, since vanilla pruning also activates a larger fraction of the expert pool. This matches the intended operating regime of OEA: moderate batch sizes where there is enough expert sharing to piggyback, but not so much saturation that pruning already activates nearly all experts.

# B. Cross entropy ablation plots

As a lot of runs present negligible differences in cross-entropy and average number of activated experts, we round the increase in cross-entropy to the closest multiple of $0.005$ and the average number of activated experts to the closest multiple

*Table 6.* Ablation across $k_0$: Benchmark accuracies, *standard error included*, of **simplified OEA** routing (top-$k_0$+piggybacking) on Qwen3-235B-A22B. Vanilla represents the default model.

|  | $k_0 = 3$ | $k_0 = 4$ | $k_0 = 5$ | $k_0 = 6$ | VANILLA |
|---|---|---|---|---|---|
| AIME24 | $81.4 \pm 1.21$ | $82.5 \pm 0.48$ | $83.6 \pm 1.39$ | $83.6 \pm 1.11$ | $85.0 \pm 0.96$ |
| GPQA | $66.3 \pm 0.67$ | $67.7 \pm 1.05$ | $67.5 \pm 0.67$ | $67.5 \pm 0.45$ | $68.4 \pm 0.34$ |
| LIVECODEBENCH | $63.4 \pm 0.75$ | $67.1 \pm 0.73$ | $66.1 \pm 0.63$ | $66.1 \pm 0.12$ | $68.5 \pm 0.21$ |
| MATH_500 | $94.4 \pm 0.46$ | $94.8 \pm 0.31$ | $94.7 \pm 0.13$ | $94.3 \pm 0.18$ | $94.7 \pm 0.18$ |

*Table 7.* Ablation across $k_0$: Benchmark accuracies, *standard error included*, of **pruned** routing (top-$k_0$) on Qwen3-235B-A22B. Vanilla represents the default model.

|  | $k_0 = 3$ | $k_0 = 4$ | $k_0 = 5$ | $k_0 = 6$ | VANILLA |
|---|---|---|---|---|---|
| AIME24 | $17.5 \pm 0.96$ | $69.4 \pm 0.73$ | $81.9 \pm 1.39$ | $82.8 \pm 1.21$ | $85.0 \pm 0.96$ |
| GPQA | $43.8 \pm 0.61$ | $56.4 \pm 0.73$ | $60.6 \pm 1.54$ | $64.1 \pm 1.01$ | $68.4 \pm 0.34$ |
| LIVECODEBENCH | $5.7 \pm 0.72$ | $27.4 \pm 1.19$ | $53.5 \pm 0.73$ | $60.8 \pm 0.24$ | $68.5 \pm 0.21$ |
| MATH_500 | $80.9 \pm 0.18$ | $93.3 \pm 0.27$ | $94.5 \pm 0.44$ | $94.5 \pm 0.18$ | $94.7 \pm 0.18$ |

of 0.1 to avoid crowding the plots.

### B.1. Pruned vs OEA ablation

We performed ablations for all batch sizes showing the Pareto frontier of OEA-based experiments in contrast to simple pruning (based on phase 1). The results are shown in Figure 4 and confirm the piggybacking's (phase 2) gains.

### B.2. Ablation over maxP

Pareto frontiers corresponding to each of the values of maxP are displayed in Figure 5 for all batch sizes $B$ and support the fact that maxP $= 128$ is optimal while maxP $= 8$ is strictly worse.

### B.3. Ablation over $k^{\mathrm{max}}$

Ablations over the value of $k^{\mathrm{max}}$ are depicted in Figure 6, with a different Pareto frontier computed for each $k^{\mathrm{max}}$. Note that $k^{\mathrm{max}} = 8$ and $k^{\mathrm{max}} = 9$ perform comparably with others being strictly worse.

### B.4. Simplified OEA contrasted with other settings

Figure 7 contrasts the Pareto frontier of simplified OEA (Algorithm 2) with all the other settings (pruned and general OEA together). It shows no meaningful trade-off losses.

### B.5. Ablation over $p$

We grouped experiments by whether $p = 1$ (thus having a static $k_0$ core experts per token) or $p < 1$, as well as whether they use a pruned (phase-1) routing or an OEA-based routing. Pareto frontiers for these 4 groups are depicted in Figure 8. Note that within both pruned and OEA, it consistently holds that $p = 1$ approximately recovers performance of $p < 1$.

## C. Hardware and model

Unless otherwise specified, all our experiments were performed on one NVIDIA H100 80GB GPU each, while using Qwen3-30B-A3B (Yang et al., 2025) under `bfloat16` precision. This model has 48 layers, each with $N = 128$ experts of which $k = 8$ are activated per token, 32 query heads and 4 KV heads, an embedding dimension of 2048 and per-expert hidden dimension of 768; each expert uses SwiGLU-based (Shazeer, 2020) feedforward network which entails 3 matrix multiplications of sizes $2048 \times 768$.

*Table 8.* Average number of activated experts when using **simplified OEA** (top-$k_0$ + piggybacking) on Qwen3-235B-A22B.

|  | $k_0 = 3$ | $k_0 = 4$ | $k_0 = 5$ | $k_0 = 6$ | VANILLA |
|---|---|---|---|---|---|
| AIME24 | 27.5 | 32.9 | 38.4 | 43.9 | 53.2 |
| GPQA | 27.4 | 33.3 | 38.6 | 43.1 | 51.6 |
| LIVECODEBENCH | 28.8 | 34.9 | 41.2 | 45.8 | 55.1 |
| MATH_500 | 29.6 | 36.2 | 42.4 | 46.2 | 56.0 |
| AVERAGE | 28.3 | 34.4 | 40.2 | 44.7 | 54.0 |
| NORMALIZED AVERAGE | 0.53 | 0.64 | 0.74 | 0.83 | 1.00 |

*Table 9.* Average MoE layer latency (in microseconds) when using **simplified OEA** (top-$k_0$ + piggybacking) on Qwen3-30B-A3B.

|  | $k_0 = 3$ | $k_0 = 4$ | $k_0 = 5$ | $k_0 = 6$ | $k_0 = 7$ | VANILLA |
|---|---|---|---|---|---|---|
| AIME24 | 97.9 | 110.5 | 122.0 | 138.4 | 147.4 | 158.0 |
| GPQA | 111.0 | 125.4 | 143.2 | 159.0 | 172.5 | 184.1 |
| LIVECODEBENCH | 102.7 | 117.1 | 132.2 | 146.5 | 157.3 | 170.8 |
| MATH_500 | 115.6 | 130.4 | 146.7 | 161.5 | 174.7 | 189.9 |
| AVERAGE | 106.8 | 120.9 | 136.0 | 151.3 | 163.0 | 175.7 |
| NORMALIZED AVERAGE | 0.61 | 0.69 | 0.77 | 0.86 | 0.93 | 1.00 |

## D. Discussion

**Effect of batch distribution.** OEA's effectiveness depends strongly on the tokens' distribution within a batch. When tokens come from similar distributions, they tend to route to overlapping experts resulting in a smaller $S^{\text{base}}$ which limits piggybacking's gains. This is the regime that our benchmarks represent, making the reported performance a conservative estimate. In contrast, the cross-entropy experiments in Section 4.1 correspond to a more diverse token distribution which enlarges $S^{\text{base}}$ and allows piggybacking to recover more of the base model's performance.

**A note on padding.** During our experiments, under the default configuration of SGLang, we noticed the average number of tokens (and average latency) in batches of size 7 exceeded that of batches of size 8, which was counter-intuitive. This is because SGLang captures CUDA Graphs for a set of batch sizes and when it needs to process a certain batch size $B$, it looks up the smallest $B' > B$ that has been captured and pads the batch up to size $B'$. While for classic feed-forward networks and attention, the specific contents of the batch do not influence the kernel's runtime, this is not the case for MoEs, especially under the memory-bound batch regime we operate. We observed that the padding token activated on average more experts "out-of-distribution" (i.e., experts not activated by real tokens) than an 8th realistic one would. Thus, this seemingly inoffensive padding incurred a higher cost than processing an extra real token, when it should ideally not add any more experts. In our experiments, we simply fixed this by capturing CUDA Graphs up to size 16 (thus ensuring no padding), but we do make the general note that there is value in adding a padding mask and using it to zero out the padding tokens' expert choices.

*Table 10.* MMLU-Redux results without thinking mode for Qwen3-30B-A3B at batch size 16. We report pruned top-$k_0$ accuracy, simplified OEA accuracy, the average number of activated experts under OEA, MoE-layer latency, and normalized MoE-layer latency.

| | PRUNED ACC. | OEA ACC. | OEA #EXPERTS | OEA LATENCY | NORMALIZED LATENCY |
|---|---|---|---|---|---|
| $k_0 = 8$ (VANILLA) | $79.4 \pm 0.18$ | $79.4 \pm 0.18$ | 68.09 | 11.42 | 1.00 |
| $k_0 = 7$ | $79.3 \pm 0.17$ | $79.5 \pm 0.21$ | 63.17 | 10.72 | 0.94 |
| $k_0 = 6$ | $79.1 \pm 0.30$ | $79.2 \pm 0.30$ | 57.07 | 9.86 | 0.86 |
| $k_0 = 5$ | $78.5 \pm 0.21$ | $79.0 \pm 0.09$ | 50.24 | 8.90 | 0.78 |
| $k_0 = 4$ | $77.4 \pm 0.36$ | $79.0 \pm 0.26$ | 43.24 | 7.91 | 0.69 |
| $k_0 = 3$ | $75.2 \pm 0.18$ | $78.6 \pm 0.23$ | 34.96 | 6.74 | 0.59 |

*Table 11.* MMLU-Redux results without thinking mode for Qwen3-235B-A22B at batch size 16. We report pruned top-$k_0$ accuracy, simplified OEA accuracy, the average number of activated experts under OEA, MoE-layer latency, and normalized MoE-layer latency.

| | PRUNED ACC. | OEA ACC. | OEA #EXPERTS | OEA LATENCY | NORMALIZED LATENCY |
|---|---|---|---|---|---|
| $k_0 = 8$ (VANILLA) | $83.3 \pm 0.35$ | $83.3 \pm 0.35$ | 68.84 | 13.64 | 1.00 |
| $k_0 = 6$ | $82.0 \pm 0.20$ | $83.1 \pm 0.20$ | 57.72 | 12.10 | 0.89 |
| $k_0 = 5$ | $81.2 \pm 0.14$ | $82.9 \pm 0.16$ | 51.06 | 11.21 | 0.82 |
| $k_0 = 4$ | $78.9 \pm 0.21$ | $82.2 \pm 0.15$ | 43.64 | 10.25 | 0.75 |
| $k_0 = 3$ | $72.9 \pm 0.46$ | $81.7 \pm 0.17$ | 35.21 | 9.32 | 0.68 |

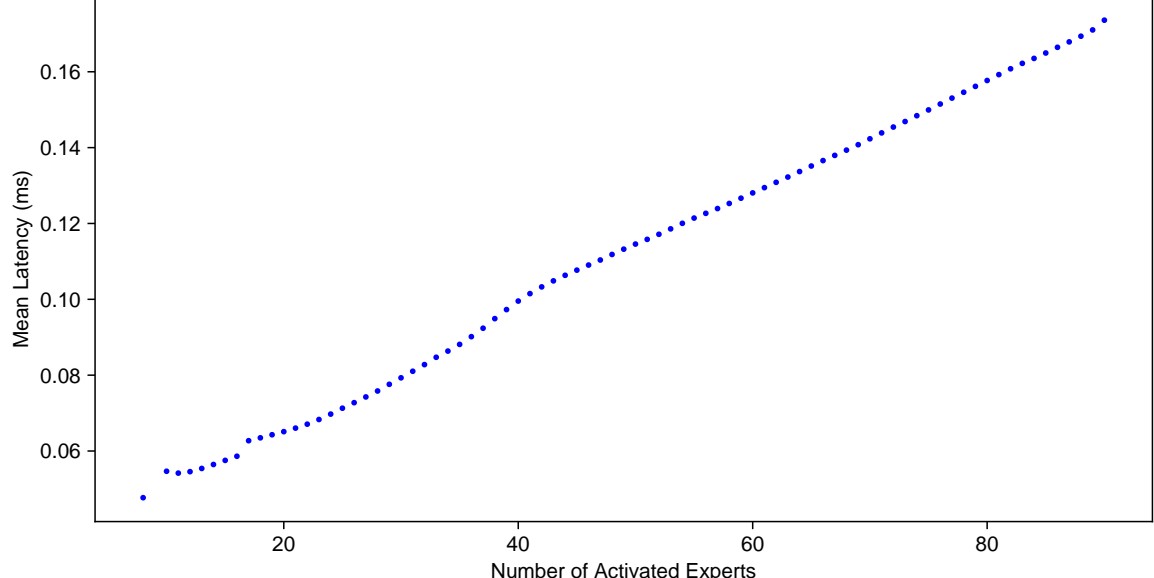

*Figure 3.* Mean MoE latency as a function of the number of activated experts within a decode batch. The average is computed over all layers and decode steps across a GPQA evaluation of the vanilla Qwen3-235B-A22B model (under a tensor parallel degree of 8).

*Table 12.* Effect of batch size on MMLU-Redux without thinking mode for Qwen3-235B-A22B. The pruned accuracy column is independent of batch size, while OEA accuracy and normalized MoE-layer latency are reported for batch sizes 8, 16, and 32.

| | PRUNED ACC. | OEA ACC. $B = 8$ | NORM. LAT. $B = 8$ | OEA ACC. $B = 16$ | NORM. LAT. $B = 16$ | OEA ACC. $B = 32$ | NORM. LAT. $B = 32$ |
|---|---|---|---|---|---|---|---|
| $k_0 = 8$ (VANILLA) | $83.3 \pm 0.35$ | – | 1.00 | – | 1.00 | – | 1.00 |
| $k_0 = 6$ | $81.7 \pm 0.14$ | $82.8 \pm 0.30$ | 0.86 | $83.1 \pm 0.20$ | 0.89 | $82.8 \pm 0.47$ | 0.92 |
| $k_0 = 5$ | $81.0 \pm 0.28$ | $82.0 \pm 0.15$ | 0.80 | $82.9 \pm 0.16$ | 0.82 | $83.1 \pm 0.22$ | 0.86 |
| $k_0 = 4$ | $79.4 \pm 0.19$ | $81.8 \pm 0.40$ | 0.72 | $82.2 \pm 0.15$ | 0.75 | $82.7 \pm 0.12$ | 0.79 |
| $k_0 = 3$ | $72.7 \pm 0.04$ | $80.7 \pm 0.15$ | 0.65 | $81.7 \pm 0.17$ | 0.68 | $82.2 \pm 0.55$ | 0.71 |

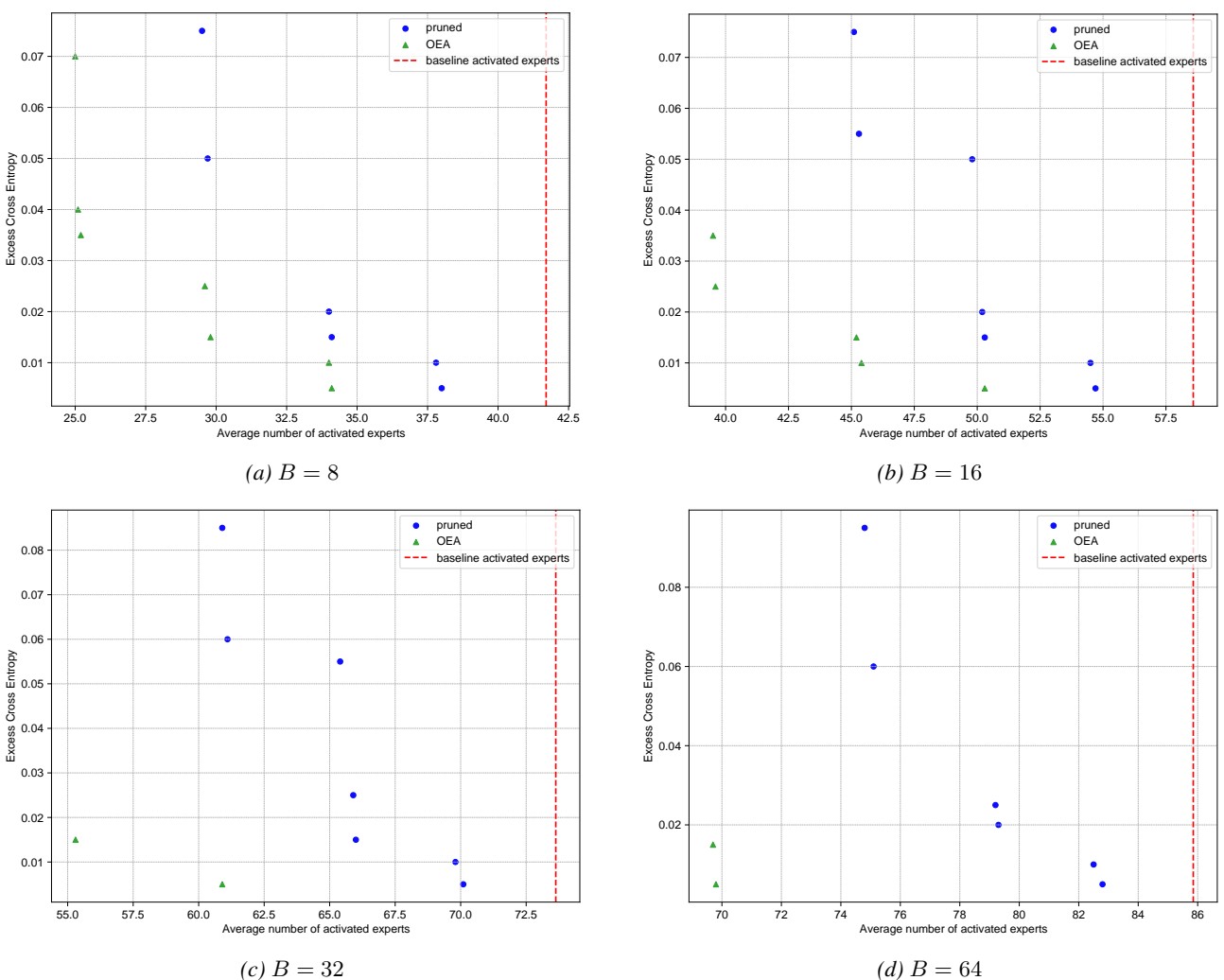

*(a) $B = 8$*

*(b) $B = 16$*

*(c) $B = 32$*

*(d) $B = 64$*

*Figure 4.* The y-axis shows the cross-entropy delta relative to the baseline (lower left is better). The two types of dots correspond to the Pareto frontiers of pruned and OEA experiments at all batch-sizes $B$. OEA consistently performs better.

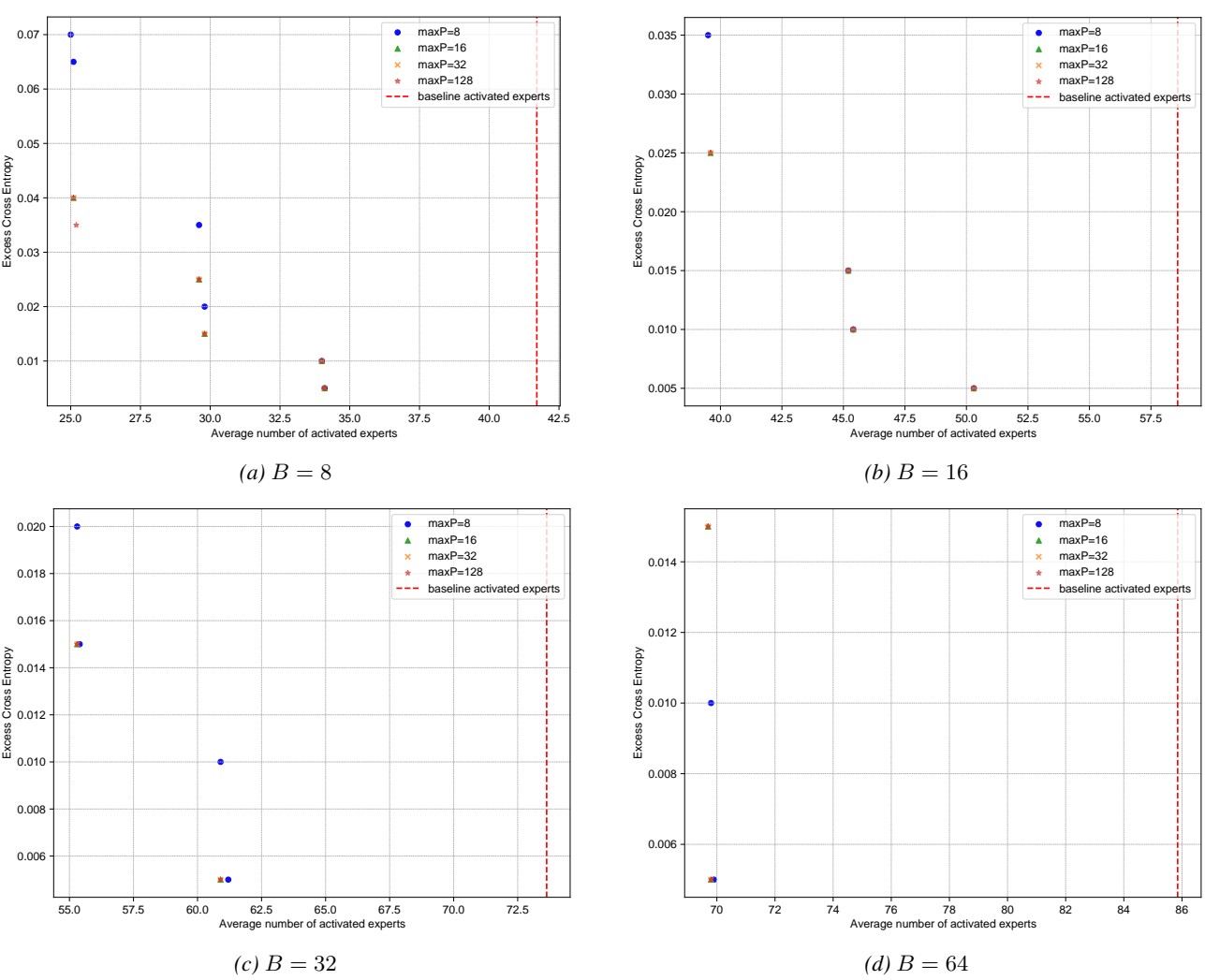

*Figure 5.* The y-axis shows the cross-entropy delta relative to the baseline (lower left is better). The four types of dots correspond to the Pareto frontiers of experiments using different values of maxP. maxP = 128 consistently performs best, whereas maxP = 8 is strictly worse than it.

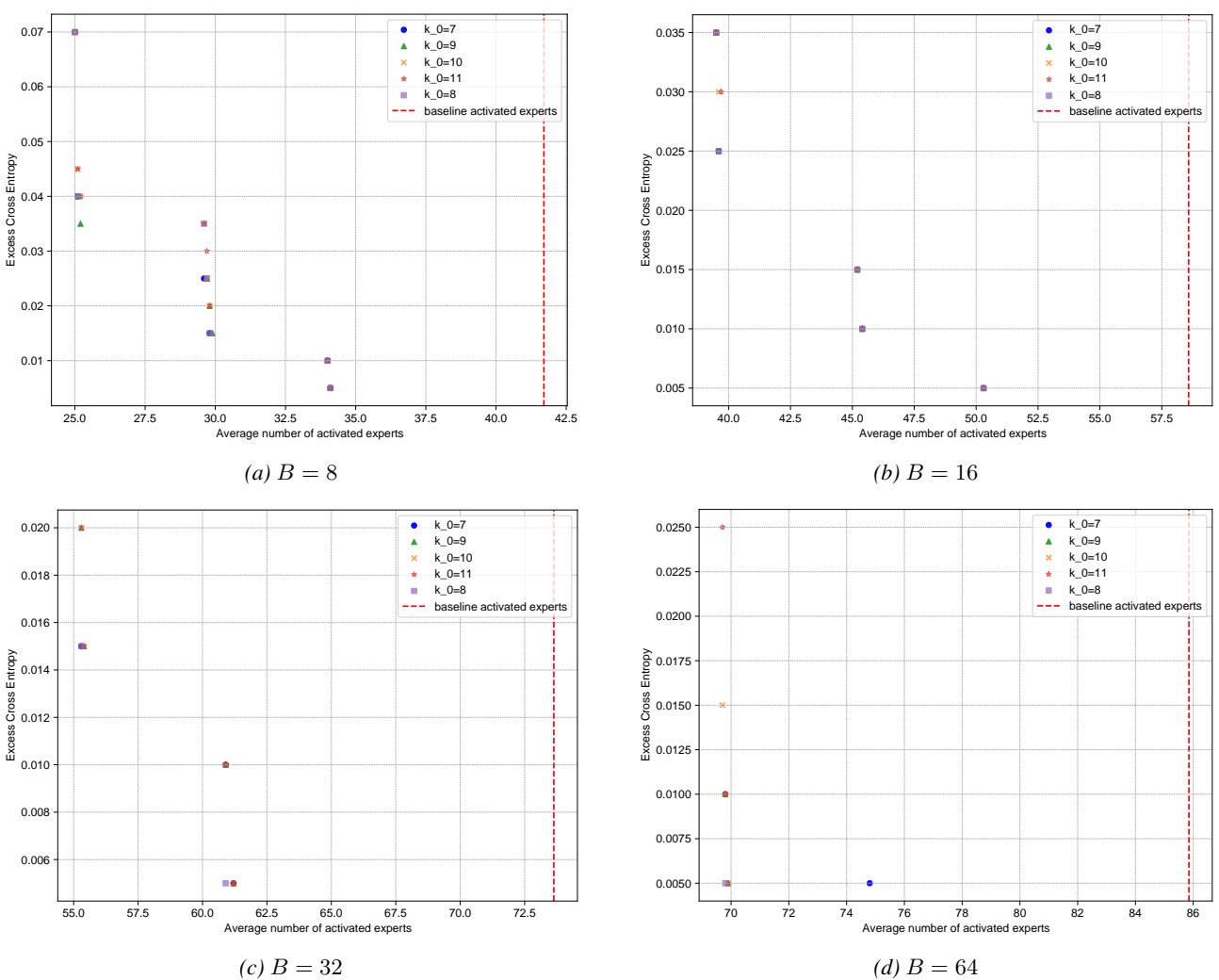

*Figure 6.* The y-axis shows the cross-entropy delta relative to the baseline (lower left is better). The five types of dots correspond to the Pareto frontiers experiments using different values of $k^{\max}$. $k^{\max} \in \{8, 9\}$ perform best while all others perform strictly worse.

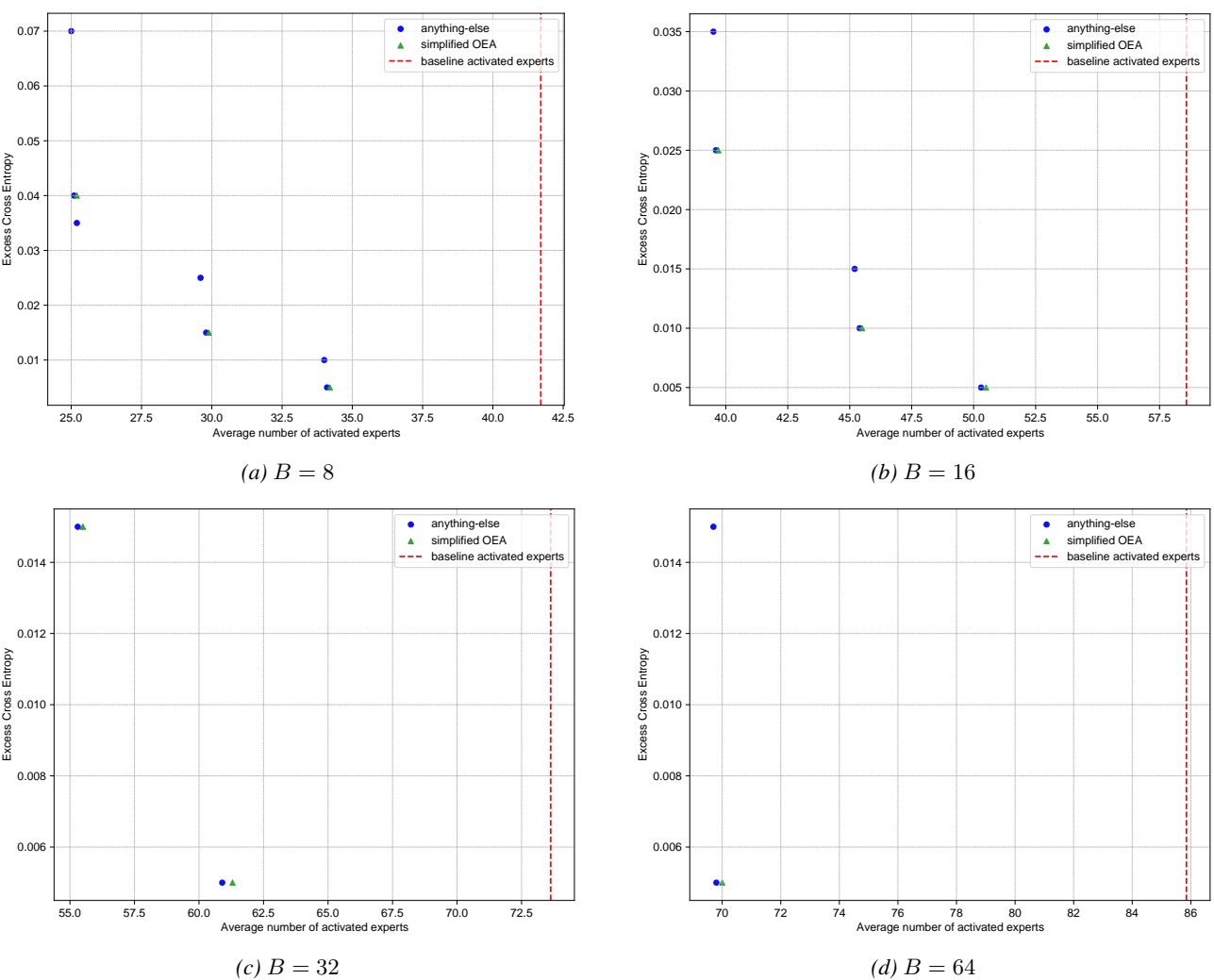

*Figure 7.* The y-axis shows the cross-entropy delta relative to the baseline (lower left is better). The two types of dots correspond to the Pareto frontiers of simplified OEA and the rest of experiments at all batch-sizes $B$. Simplified OEA performs comparably to the best hyperparameter choices.

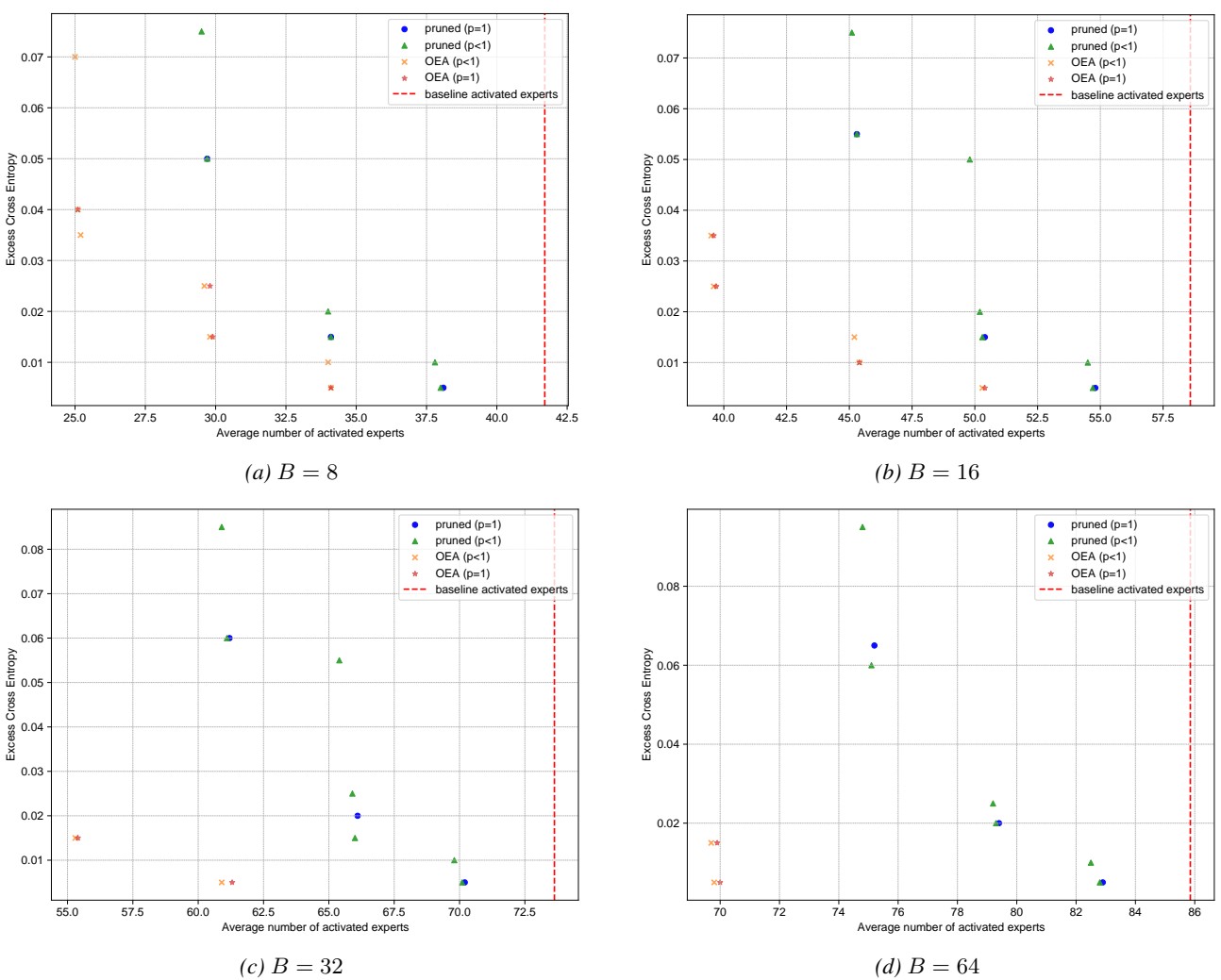

*Figure 8.* The y-axis shows the cross-entropy delta relative to the baseline (lower left is better). We split dots as per the legend (by whether $p = 1$ and whether they use pruned or OEA-based routings) and report the Pareto frontiers of each group for all batch sizes $B$. Always using $p = 1$ does not compromise substantial performance within either group.

