# OpenReview forum: "Opportunistic Expert Activation: Batch-Aware Expert Routing for Faster Decode Without Retraining"
_ICML.cc/2026/Conference — ICML 2026 regular_

### Official Review · Reviewer_9Pc7 · 2026-02-16

**Soundness:** 4
**Presentation:** 3
**Significance:** 2
**Originality:** 3
**Overall Recommendation:** 4
**Confidence:** 5

**Summary:**

Since Mixture-of-Experts (MoE) models replace the feed-forward (FF) network in the transformer block with multiple FF networks, a batch is fragmented into smaller sub-batches and routed to different experts, further decreasing arithmetic intensity.

This paper provides a serving approach that aims to reduce the number of activated experts, thereby reducing both the quality of the generated output and the latency of the decode step. The first part of the approach prunes the required experts for every token in the batch (e.g., using 3 instead of 8). The second part checks for already included experts to find piggybacking opportunities to compensate for the lost quality of the generated output.

**Compliance With Llm Reviewing Policy:**

Affirmed.

**Final Justification:**

I maintain my positive score.

**Key Questions For Authors:**

- How would the performance gains change when using this approach on other MoE architectures that have a different number of experts per layer and different activation sparsity?
- How do you explain the reduction in benchmark accuracies when the second part of the approach is added to the pruning?
- After a certain batch size, by only applying pruning, all of the experts in a layer would be required. Can you relate that maximum batch size to the architecture of an MoE model?
- Although your approach imposes simple computations as overhead, do you have any estimate of the added latency?

**Limitations:**

Yes.

**Strengths And Weaknesses:**

### Strengths:

- The proposed approach is well-defined and well-presented.
- The results and experiments are technically sound.
- The paper aims to address the challenges of MoE inference, which is the current SOTA architecture for LLMs.
- The second part of the proposed approach is original and adds to the prior research.

### Weaknesses:

- The ablation studies and experiments are conducted using two model architectures (Qwen3-30B-A3B and Qwen3-235B-A22B) that have the same number of experts per layer and activation sparsity (8 out of 128).

- As mentioned by the authors in Section 4.2 (Line 258, "Piggybacking gains") of the paper, the gains from the second part of the approach-which is the main contribution-are limited to $k_0 < 6$.

- As shown in the results section, the second part of the approach makes the quality of the generated output even worse in some cases.

- There is no discussion about the overheads imposed by this serving approach.

##

---

> ### Author Rebuttal · Authors · 2026-03-31
>
> Thank you for the thoughtful review and for acknowledging our contributions. We answer your questions and comments below.
>
> - Regarding Q1/W1 (and also Q3):
>   This is a very good point, and we are happy to discuss it more explicitly in the paper. It is not a coincidence that we chose a model with top-8 out of 128 experts. Our method is only useful as long as not all experts are already activated, so if the sparsity is low, one should expect even small batch sizes to already require most or all experts, leaving little room for OEA to help. This also depends on $k_0$: if $k$ is small enough that no $k_0 < k$ preserves a good baseline quality, then the method would naturally be ineffective. As a practical takeaway, OEA is best suited to high-sparsity, relatively high-$k$ MoE models.
>   We can also formalize this under a uniform-routing assumption. Let $B_{\min}$ denote the minimum batch size required to activate all experts under top-$k$ routing with $N$ experts. Then
>   $$
>   \mathbb{E}[B_{\min}] \geq \frac{NH_N}{k} \approx \frac{N}{k}\log N,
>   $$
>   where $H_N$ is the $N$-th harmonic number. This shows that the sparsity ratio $N/k$ lower-bounds how quickly we reach the regime where all experts are covered. For OEA, the same intuition applies with $k_0$ in place of $k$. We are happy to add this discussion, and potentially a small simulation as well, to better illustrate how the speedups depend on $N, k, k_0,$ and $B$.
>
> - Regarding Q2/W3:
>   Our interpretation is that these cases are mostly noise, or possibly a mild regularization effect. The only example with a larger gap is $k_0 = 7$ on AIME24 for the 30B model. Tables 4 and 5 show that the standard deviations there are still reasonably large ($0.83$ and $1.33$). More importantly, AIME24 has only 30 questions, so each question accounts for roughly 3\% of the score; this makes small differences quite sensitive to individual problems. It could therefore simply reflect a particular prompt where using all 8 experts is not optimal (and, notably, vanilla is also lower there).
>
> - Regarding Q4/W4:
>   We agree that the routing overhead should be discussed more clearly. At the moment, we do not have an optimized kernel for the routing itself, which is why it is not included in the profiling; the current analysis is mainly about the theoretical limit of the approach. That said, the routing has a useful systems property: it only operates on a $B \times E$ tensor, which can easily fit in SRAM on a modern GPU and could in principle be handled in a single fused kernel. For this reason, we expect that an optimized implementation would add only a negligible overhead.
>
> - Regarding W2:
>   It is true that, on the current benchmarks, the clearest gains from piggybacking appear for $k_0 < 6$. However, this is mainly because naive pruning already matches vanilla quite well at milder pruning levels. In practice, benchmark coverage is always limited, and one could still encounter serving regimes where, for example, $k_0 = 7$ performs worse than suggested by the current suite. OEA may not always be enough to recover such cases completely, but it also comes with essentially no additional computational cost beyond pruning itself, assuming an efficient router implementation. In that sense, even at mild pruning levels it can still serve as an additional degree of quality protection.

---

> > ### Author Rebuttal · Reviewer_9Pc7 · 2026-04-03
> >
> > I appreciate your rebuttal and further clarification.
> >
> > My concerns are mostly resolved.
> >
> > I will keep my positive score.

---

### Official Review · Reviewer_vsam · 2026-02-21

**Soundness:** 4
**Presentation:** 4
**Significance:** 3
**Originality:** 3
**Overall Recommendation:** 5
**Confidence:** 4

**Summary:**

I believe this paper is overall very excellent. It achieves improvements while being training-free, and on Qwen3-30B-A3B it both improves performance and increases TPS. I consider the results to be solid. Moreover, the validation on large-scale MoE models ensures its applicability and effectiveness in industrial scenarios.

**Compliance With Llm Reviewing Policy:**

Affirmed.

**Key Questions For Authors:**

1. Which version of LiveCodeBench is used, v5 or v6? This should be specified.

2. Would similar issues occur on other inference GPUs, such as H200 or H20-3E?

3. For MoE models with shared experts, do the authors believe that adapting text-based methods would bring larger gains?

**Limitations:**

See Weakness.

**Strengths And Weaknesses:**

Other Strengths include:

1. Although the number of evaluation datasets is not large, each of them is relatively challenging and representative.

2. The motivation is clear and rigorous, and the paper provides relatively complete descriptions of important details.

3. The testing adapts to sglang, which aligns with real-world application scenarios.

Weaknesses:

1. The number of evaluation datasets is somewhat limited (mainly concentrated on high-difficulty code, math, and reasoning datasets). Since these datasets are challenging, the average number of output tokens during decoding is relatively large, which may potentially overestimate the acceleration effect. It is recommended to include more general knowledge datasets such as MMLU, or simpler ones (e.g., MultiPL-E, BFCL, IFEval).

2. The experiments only use the Qwen-3 series. Although I do not consider this a major issue, testing on the Qwen3-2507 series or other MoE models would make the results more convincing.

---

> ### Author Rebuttal · Authors · 2026-03-31
>
> Thank you for your review and for acknowledging the potential of the work! We answer your questions below.
>
> - We used LiveCodeBench **v5** and we will update the paper to state this explicitly.
>
> - Regarding whether similar issues would arise on other inference hardware: yes, we expect the same qualitative effect whenever the hardware has a large ratio between compute bandwidth and memory bandwidth, since this determines the arithmetic intensity required to leave the memory-bound regime and thus the critical batch size beyond which experts become compute-bound. We are happy to add a short discussion explaining why the analysis and latency model are hardware-agnostic in this sense, and why the same issue can persist on newer GPUs as well.
>
> - Regarding models with shared experts: while we did not test OEA in that setting, we would expect applying it to the routed experts to have a similar effect. The main difference is that the latency model would then include an additional term linear in $B$ coming from the shared experts, but we would not expect this term to dominate since its loading cost is constant across tokens.
>
> - To address the evaluation weakness, we also ran MMLU-Redux on both the 30B and 235B models. One point worth clarifying is that the speedups reported in the paper are per decode token in the MoE part, precisely so as to isolate the effect of routing from the number of decode steps and the context length.
>
> For consistency, we ran MMLU-Redux without thinking mode on both models, still at batch size 16. Since the chains of thought are much shorter in this setting, even larger batch sizes would have been feasible, but batch size 16 is already enough to recover most of vanilla performance. The corresponding results are shown below. The 30B model sees some mild deteriorations at $k_0=3$ (of less than $0.8\\%$) with even larger speed-ups than before, whereas $k_0=5$ for the 235B model achieves a speed-up of $18\\%$ while not performing standard-error adjusted worse than vanilla. This is while improving upon naive pruning at $k_0=3$ by $9\\%$.
>
> ### Qwen3-30B-A3B:
> | k | pruned acc | OEA acc | OEA #experts | OEA latency | normalized latency |
> | --- | --- | --- | --- | --- | --- |
> | $k=8$ (vanilla) | 79.4 ± 0.18 | (same) | 68.09 | 11.42 | 1.00 |
> | $k_0=7$ | 79.3 ± 0.17 | 79.5 ± 0.21 | 63.17 | 10.72 | 0.94 |
> | $k_0=6$ | 79.1 ± 0.30 | 79.2 ± 0.30 | 57.07 | 9.86 | 0.86 |
> | $k_0=5$ | 78.5 ± 0.21 | 79.0 ± 0.09 | 50.24 | 8.90 | 0.78 |
> | $k_0=4$ | 77.4 ± 0.36 | 79.0 ± 0.26 | 43.24 | 7.91 | 0.69 |
> | $k_0=3$ | 75.2 ± 0.18 | 78.6 ± 0.23 | 34.96 | 6.74 | 0.59 |
>
>
>
> ### Qwen3-235B-A22B:
> | k | pruned acc | OEA acc | OEA #experts | OEA latency | normalized latency |
> | --- | --- | --- | --- | --- | --- |
> | k=8 (vanilla) | 83.3 ± 0.35 | (same) | 68.84 | 13.64 | 1.00 |
> | k_0=6 | 82.0 ± 0.20 | 83.1 ± 0.20 | 57.72 | 12.10 | 0.89 |
> | k_0=5 | 81.2 ± 0.14 | 82.9 ± 0.16 | 51.06 | 11.21 | 0.82 |
> | k_0=4 | 78.9 ± 0.21 | 82.2 ± 0.15 | 43.64 | 10.25 | 0.75 |
> | k_0=3 | 72.9 ± 0.46 | 81.7 ± 0.17 | 35.21 | 9.32 | 0.68 |

---

> > ### Author Rebuttal · Reviewer_vsam · 2026-04-04
> >
> > l appreciate the rebuttal.

---

### Official Review · Reviewer_D8pW · 2026-03-12

**Soundness:** 2
**Presentation:** 2
**Significance:** 2
**Originality:** 3
**Overall Recommendation:** 3
**Confidence:** 4

**Summary:**

The paper studies decode-time latency of Mixture-of-Experts (MoE) LLMs in the memory‑bound regime and argues that latency is effectively governed by the number of distinct experts activated per batch rather than FLOPs. Building on a simple linear latency model  and empirical measurements, the authors propose Opportunistic Expert Activation, a batch‑aware routing scheme that keeps a token‑wise baseline of top experts and then "piggybacks" additional experts only if they are already active for other tokens in the batch. Experiments on Qwen3‑30B and Qwen3‑235B show sizeable reductions in active experts and MoE-layer latency with minimal degradation in cross‑entropy and downstream accuracy.

**Compliance With Llm Reviewing Policy:**

Affirmed.

**Final Justification:**

I maintain my score at weak reject. The rebuttal provides useful clarifications and acknowledges some issues (e.g., figure errors and related work), but my main concerns are not fully resolved. In particular, the evaluation remains limited, and the evidence does not yet fully support the strength of the paper’s claims and comparisons.

**Key Questions For Authors:**

1. Why do the two types of data points in Figure 2 have different counts? How to determine which points correspond to the same setting, or are the two types of points not intended to be paired?
2. In Figure 2(b), why can the performance of Simplified OEA exceed Standard routing? This seems somewhat inconsistent with the results in Table 2 and Table 3.
3. Table 1 should include results on more benchmarks.
4. How is OEA affected by batch size? Under different batch sizes, is there any performance degradation or increase in latency?

**Limitations:**

yes

**Strengths And Weaknesses:**

Strengths
1. Clear problem formulation and latency model.
2. Meaningful latency and accuracy results on large, modern MoE LLMs.

Weaknesses

1. Limited evaluation regime. Only four benchmarks are considered and somewhat fragile narrative around "no statistically significant degradation". The effectiveness of this method may depend on certain distributional characteristics within the benchmarks, and it is unclear how it would perform on other benchmarks such as MMLU.

2. The comparison underplays the breadth of the existing literature. several methods share closely overlapping goals with OEA.

[1] SERE: Similarity-based Expert Re-routing for Efficient Batch Decoding in MoE Models. ICLR 2026

[2] Efficient MoE Serving in the Memory-Bound Regime: Balance Activated Experts, Not Tokens.

[3] eMoE: Task-aware Memory Efficient MoE Inference

3. Some experimental settings are not clearly described.

---

> ### Author Rebuttal · Authors · 2026-03-31
>
> Thank you for your thoughtful questions and for the helpful references! We answer the main points below.
>
> - Regarding Q1:
>   Figure 2 reports Pareto frontiers of settings of different kinds: in (a), the split is between pruned routing and OEA, while in (b), the split is between simplified OEA and the general OEA settings. The points are therefore not meant to admit a 1-1 correspondence. Even beyond the fact that we only display the Pareto frontiers, there would still not be a natural pairing: for example, a single pruned setting can underlie several OEA variants depending on $k^{\max}$ and $\mathrm{maxP}$. The intended comparison is between the frontiers themselves, i.e. between the efficiency / cross-entropy compromises achievable by each family of methods.
>
> - Regarding Q2:
>   You are absolutely right; this was an unfortunate legend error. Simplified OEA should not exceed the frontier of the full set of methods. The legend was meant to say **“everything else”**, as in Figure 7 of the appendix; in fact, the main-text figure is just one of the four appendix frames enlarged for readability. The point of that figure is to show that restricting to simplified OEA leaves the Pareto frontier, and thus the attainable tradeoffs, essentially unchanged, which is exactly the conclusion of the “Simplifying OEA” paragraph (lines 299-308). We will fix the legend, and thank you for catching this.
>
> - Regarding Q4:
>   We are happy to make the batch-size interplay more explicit. For a fixed $k_0$, as batch size increases, OEA’s accuracy should improve, up to matching vanilla once all experts become activated. The speedup is then governed mostly by the ratio between the number of activated experts, which can be computed in expectancy via the formula from footnote 1 - it will drop with batch size until it converges to 1. Our cross-entropy experiments already cover a broader range of batch sizes and support this picture. The reason we could not push the downstream decode experiments to larger batch sizes is that the large number of decode steps (up to 32K) makes the KV cache too large.
>
> - Regarding the weaknesses on evaluation and distributional dependence and Q3:
>   We agree that the effectiveness depends on the distribution within the batch. The cross-entropy experiments are meant to cover a much more diverse distribution, while also allowing us to vary batch size and hyperparameters much more extensively and cheaply. Following your suggestion, we also ran **MMLU-redux** on both model sizes, and these additional results support the same overall conclusions. Please see our response to Reviewer vsam, where we include these results.
>
> - Regarding the related-work suggestions:
>   The first two references are indeed very relevant, and we are grateful for the heads-up. They are concurrent with our submission - the first appeared after the submission deadline, while the second was posted to arXiv less than two months before the deadline - which is why they were not included originally. We agree they should be discussed in the revision. eMoE also fits very naturally into the expert-prefetching discussion, and we are happy to add it there together with related systems such as EdgeMoE and MoEInfinity.
>
> - Regarding the missing experimental details:
>   Thank you for pointing this out. We will specify the LiveCodeBench version, as also suggested by Reviewer vsam. If there are additional settings you felt were missing, we would be very happy to clarify those as well.

---

> > ### Author Rebuttal · Reviewer_D8pW · 2026-04-04
> >
> > I appreciate the authors' clarifications, including acknowledging the legend error in Figure 2 and committing to include the relevant concurrent/prior works. However, my concerns regarding the empirical evaluation and the paper's central claims remain unresolved.
> >
> > 1. The claim of "without any statistically significant loss in accuracy" is an overclaim, especially given the clear performance drop shown in Table 1(b). While the authors provided one additional benchmark during the rebuttal, it remains unclear whether the method would suffer more performance degradation on other standard benchmarks(e.g. https://huggingface.co/spaces/open-llm-leaderboard/open_llm_leaderboard#/). Since this is an inference-only method, the computational cost for evaluation is relatively low. The authors should provide a much broader suite of evaluation results to convincingly support their claims.
> >
> > 2. For a paper targeting batch-aware routing, it is critical to evaluate how downstream accuracy and latency vary across different common batch sizes.
> >
> > 3. Regarding the clarification of Figure 2, because OEA has a significantly larger hyperparameter space compared to the baseline comparing their empirical Pareto frontiers without controlling the search space introduces an unfair hyperparameter tuning bias.

---

> > > ### Author Response · Authors · 2026-04-07
> > >
> > > Thank you for the follow-up comments.
> > >
> > > - We agree that the original phrasing was too strong, and we will revise it. For the 235B model, it is more accurate to say that OEA incurs small degradation overall rather than “without any statistically significant loss in accuracy.”
> > >
> > > - On the evaluation breadth: while OEA is an inference-only method, the evaluations in the regime we study are still more costly than they may appear at first glance. The main reasons are that the downstream experiments involve very long decode traces (often up to 32K generated tokens), large models, and repeated evaluation across multiple routing settings and runs — in our case, 9-11 configurations (varying $k_0$) and 3-5 repetitions. Moreover, decode in this regime is largely memory-bandwidth bound rather than compute-bound, and the moderate batch sizes relevant for OEA imply poor compute utilization by construction. For the 235B model, this amounts to roughly 9-10 days on a full 8xH100 node for the current downstream suite, or about 340 GPU-hours per benchmark on average.
> > >
> > > - We chose these long-thinking benchmarks intentionally. Since OEA changes routing at each decode step, long decode is where any quality loss has the most opportunity to accumulate, so this is the most important regime to test. This increased cost is also why we separately included the cross-entropy experiments, where we could study many more settings much more cheaply.
> > >
> > > - We do agree, though, that a batch-aware routing method should be evaluated across different batch sizes. To address this directly, we additionally evaluated MMLU on Qwen3-235B at batch sizes 8 and 32, in addition to 16. MMLU makes this feasible because the generations are much shorter, so larger decode batches fit comfortably (KV cache-wise). These results are included in our response to Reviewer hN2v and show the same overall trend: as batch size increases, OEA recovers more accuracy, while the relative latency gain decreases.
> > >
> > > - More broadly, this is also consistent with our cross-entropy experiments, where we already sweep over multiple values of $B$. The reason the original downstream benchmarks were run at batch size 16 is therefore not that other batch sizes are unimportant, but that 16 is the largest sustained decode batch that remains practical in the original long-thinking setting.
> > >
> > > - Regarding the Figure 2 concern: this is a fair point in the general case. The relevant subtlety here is that the paper also shows simplified OEA to have essentially the same Pareto frontier as the full OEA family. This means that the meaningful comparison can be interpreted as being between simplified OEA and pruned routing, and that comparison is fair in terms of search space since both have exactly the same one knob, namely $k_0$. We are happy to make this more explicit in the paper, or to revise the figure to show simplified OEA directly against pruned routing.
> > >
> > > We hope the additional experiments and clarifications help resolve the main concerns, and we would be grateful if you would consider them in your final review.

---

### Official Review · Reviewer_hN2v · 2026-03-19

**Soundness:** 3
**Presentation:** 2
**Significance:** 2
**Originality:** 3
**Overall Recommendation:** 3
**Confidence:** 4

**Summary:**

This paper targets the decoding phase (next token prediction) in MoE inference, where latency grows linearly with the number of unique activated experts under a moderate batch size of 16.
- The authors propose OEA, a two-phase inference-time routing algorithm. Phase 1 keeps the top-$k_0$ experts per token as a quality floor. Phase 2 re-route the tokens to reuse the experts already loaded in Phase 1.
- The authors use a roofline model to confirm the linear relationship between latency and the number of unique activated experts.
- The authors report MoE layer latency reductions of 39% (30B) and 15% (235B) at batch size 16.
- While OEA is not an exact algorithm, the authors claim that it does not cause any statistically significant loss in accuracy.

**Compliance With Llm Reviewing Policy:**

Affirmed.

**Final Justification:**

My recommendation is weak reject (3) with a confidence of 4. The rebuttal partially addressed my main concerns. The authors acknowledged the overclaim for 235B and provided results on different batch sizes. These has been factored in the current recommendation. However, the approximate nature and hyperparameter sensitivity of the method may limit practical adoption.

**Key Questions For Authors:**

I also raised some questions in Weaknesses above. I welcome but do not expect the authors to respond to every question. I am open to adjusting my scores depending on the answers/clarifications by the authors. Especially to assess whether this is a contribution that others are likely to build on.
- The abstract says "without any statistically significant loss in accuracy," but Table 1 (b) shows a clear drop. Can you reconcile this?
- In Table 1 caption, I understand that the results are averaged across 3 runs, but what does "standard-error adjusted" mean in the last sentence?
- How common are batch sizes around 16 in production or personal serving? Can you provide evidence from public reports?

**Limitations:**

Limitations are not explicitly discussed. I recommend the authors to explicitly discuss the known limitations of OEA and which scenarios are OEA most applicable. Especially that OEA relies on a large enough batch size, and that it can cause performance degradation even when $k_0$ is 6 out of 8.

**Strengths And Weaknesses:**

## Strengths
- (Significance) The 39% and 15% latency reductions are valuable. This is assuming that the model quality claim holds (see weaknesses).
- (Soundness) The authors use a roofline model to identify the primary optimization target. This is good engineering practice. Please note that the same conclusion (# of activated experts is the bottleneck) was also reported in Lynx (Gupta et al., 2024).
- (Presentation) The research problem is clearly stated. The paper is well written.

## Weaknesses
- (Important) In abstract, the authors claim that, without *any* statistically significant loss in accuracy, their approach achieves latency reductions of 39% and 15% in the MoE layer decode latency, respectively. This implies that both "Qwen3-30B" and "Qwen3-235B" do not have any statistically significant loss in accuracy. However, in Page 6, right column bottom, only "Qwen3-30B" is claimed to "not make a statistically significant difference", while "Qwen3-235B" only "maintains performance". In Table 1 (b), VANILLA consistently outperforms OEA on almost all tasks. Not uncommon to see more than 2 percentage points decrease. This *is* a statistically significant difference.
- (Presentation) In Table 1 caption, it says "Setups that are no worse than vanilla (standard-error adjusted) are in bold." However, this is not true in the actual tables. For example, in Table 1 (b) AIME24, $k_0=6$, OEA is 83.6 while VANILLA is 85.0. Also, it is unclear what "standard-error adjusted" means in Table 1 caption.
- (Significance) Piggybacking is ineffective at batch size of 1, which is common in personal serving of LLMs. This method targets moderate batch sizes, but the paper does not justify how common a batch size 16 is in real serving workloads. Does this work mainly target large scale production serving? Is a batch size of 16 representative of that scenario?
- (minor) (Significance) The method is a quality-performance tradeoff. This limits the strength of the contribution compared to an approach that preserves outputs exactly.
- (minor) (Soundness) There is no direct experimental comparison with Lynx (Gupta et al., 2024), a closely related method. The authors argue the two methods have different control knobs, but a Pareto frontier comparison similar to Figure 2 could be applied to Lynx.

---

> ### Author Rebuttal · Authors · 2026-03-31
>
> Thank you for the comprehensive review and for highlighting both the strengths of the work and the places where the presentation should be clearer! We respond to the main points below.
>
> - Regarding the first two weaknesses and the first two questions:
>   We defined what we meant by “statistical significance” in footnote 3 (page 6), and the appendix tables include the corresponding standard errors. That said, we agree that this should have been made easier to spot rather than only being defined in the footnote.
>   In particular, the bolded entries are consistent with that definition. For example, Table 6 (Appendix) shows that for Qwen3-235B on AIME24 at $k_0 = 6$, OEA gives $83.6 \pm 1.11$ while vanilla gives $85.0 \pm 0.96$; this is why the former is still bolded under the criterion used in the paper  (since $83.6 + 1.11 > 85 - 0.96$).
>   We do agree, however, that the abstract is phrased too strongly for the 235B model. For the 30B model, the conclusions are essentially unchanged, but for 235B it is more accurate to say that OEA incurs only small degradation overall rather than no statistically significant degradation. We are happy to revise this wording.
>   More broadly, because these evaluations are expensive and involve only a small number of runs, the variance is not negligible, especially for AIME24, which has only 30 questions and thus higher granularity.
>
> - Regarding the third weakness and third question:
>   Yes, piggybacking is ineffective at batch size 1 and generally becomes more useful as batch size increases. Our main target regime is moderate-batch serving, where batch size 16 is a meaningful operating point in practice [1,2]. At the same time, personal serving could also make sense whenever requests are dynamically batched, so we would not view the method as exclusive to only one deployment setting.
>   As a side note, we would in fact expect the piggybacking gains to become even stronger at somewhat larger batch sizes. In our downstream decode experiments, the main limitation was that the KV cache becomes too large because generations go up to 32K tokens, so larger decode batches were not sustainable. The accuracy should only increase with larger batch size, while the number of experts is not even close to saturation.
>
> - As for the rest of the weaknesses:
>   It is true that, since we do not preserve the exact model outputs, OEA exposes a quality-performance knob through $k_0$. In particular, in adversarial situations (e.g. batch size 1, or perfectly overlapping routing preferences across all tokens in the batch), the method can reduce to the performance of naive pruning.
>   One thing we would emphasize, though, is that by design the pruning stage provides a per-token quality lower bound, which is different from Lynx. One could in principle also set a lower bound on the batch sizes where our method is applied (we are trying to run these before the end of the discussions)
>   As for the Lynx comparison, we agree that a Pareto-style comparison would be interesting in principle, even though the knobs are not directly the same and changing data distributions could impact the comparison in unpredictable ways. Unfortunately, Lynx code has not been released, and some implementation details are ambiguous from the paper alone, so we are not able to provide a reliable comparison.
>
> Finally, we are happy to add a limitations section that covers a discussion on OEA's reliance on batch size and data distribution, including a more thorough explanation of when the method is suitable (as mentioned to Reviewer 9Pc7's point on model sparsity and setup).
>
> [1] https://arxiv.org/pdf/2411.00136
> [2] https://pytorch.org/blog/accelerating-llm-inference/

---

> > ### Author Rebuttal · Reviewer_hN2v · 2026-04-04
> >
> > The reviewer appreciate and acknowledge the authors' response. The response will be taken into consideration in the final decision.
> >
> > - "for 235B it is more accurate to say that OEA incurs only small degradation overall rather than no statistically significant degradation" Yes, this limitation should be made clear in the paper. "no statistically significant degradation" is a very strong claim.
> > - "piggybacking ... generally becomes more useful as batch size increases", it would be helpful if there are experimental results on a different batch size, since batch size is an important factor that would change how OEA behaves, such as 32 as mentioned in [2], or 1, 16, 32, 64 as mentioned in [1].
> >
> > [1] https://arxiv.org/pdf/2411.00136
> >
> > [2] https://pytorch.org/blog/accelerating-llm-inference/

---

> > > ### Author Response · Authors · 2026-04-07
> > >
> > > Thank you again for the follow-up comments! We agree with the first point, and as mentioned in our initial response, we will make that limitation explicit in the paper.
> > >
> > > - Regarding the request for additional batch-size results: the reason we did not include larger batch sizes than 16 in the original downstream experiments is that those tasks use very long chains of thought in thinking mode (often averaging over 20K generated tokens per sample). In that regime, with bfloat16 serving, batch size 32 is not sustainable for most of the decode process because the KV cache becomes too large; in practice, one cannot maintain that batch size through a majority of the decode steps for either Qwen3-30B on one GPU or Qwen3-235B on 8 GPUs.
> > >   At the other end, batch size 1 does not use piggybacking at all and is therefore exactly equivalent to the pruned baseline, so that regime is already covered.
> > >
> > > - To address your concern directly, we additionally ran MMLU on Qwen3-235B at batch sizes **8** and **32** as well. This setup makes such a comparison feasible because the generations are much shorter (around 5K tokens per prompt on average), so the KV-cache constraint is much less severe. The results are below:
> > >
> > > | k | pruned acc | OEA acc @bs=8 | norm lat @bs=8 | OEA acc @bs=16 | norm lat @bs=16 | OEA acc @bs=32 | norm lat @bs=32 |
> > > | --- | --- | --- | --- | --- | --- | --- | --- |
> > > | $k_0=8$ (vanilla) | 83.3 ± 0.35 | — | 1.00 | — | 1.00 | — | 1.00 |
> > > | $k_0=6$ | 81.7 ± 0.14 | 82.8 ± 0.30 | 0.86 | 83.1 ± 0.20 | 0.89 | 82.8 ± 0.47 | 0.92 |
> > > | $k_0=5$ | 81.0 ± 0.28 | 82.0 ± 0.15 | 0.80 | 82.9 ± 0.16 | 0.82 | 83.1 ± 0.22 | 0.86 |
> > > | $k_0=4$ | 79.4 ± 0.19 | 81.8 ± 0.40 | 0.72 | 82.2 ± 0.15 | 0.75 | 82.7 ± 0.12 | 0.79 |
> > > | $k_0=3$ | 72.7 ± 0.04 | 80.7 ± 0.15 | 0.65 | 81.7 ± 0.17 | 0.68 | 82.2 ± 0.55 | 0.71 |
> > >
> > > - As expected, the quality recovery improves gradually with batch size, while the latency gain decreases somewhat because the union of baseline experts grows with $B$. We are happy to add this discussion to the paper.
> > >
> > > - In principle, one could also run the batch-size-8 version of the original long-CoT benchmarks, but these experiments are substantially more expensive because they still require very long decode traces and repeated runs on very large models (batch size $8$ also yields an even worse hardware utilization). This is also why we included the cross-entropy experiments in the first place: they allow us to study the batch-size dependence much more broadly and cheaply, while preserving the same routing logic.
> > >
> > > We hope these clarifications and additional results address your concerns, and we would be grateful if you would take them into account in your final assessment.

---

### Decision · Program_Chairs · 2026-04-30

**Decision:**

Accept (regular)

**Comment:**

In this paper, the authors introduce Opportunistic Expert Activation (OEA), a batch-aware routing framework that reduces MoE decoding latency by limiting the number of distinct experts activated per batch during inference. The core idea is a two-phase routing scheme that reduces latency by encouraging expert sharing across tokens within a batch and preserves model quality by retaining the most important experts for each token. More specifically, the first phase keeps a small set of top-ranked experts for each token and the second phase supplements this set by selecting from experts that are already active in the batch, which then avoids additional data movement and reduces communication overhead.

The method is siginificant, novel and straightforward to apply in practice. With the growing demand for low-latency inference in large-scale models, the reported latency reductions over the vanilla model are meaningful. Additionally, the method also preserves expert diversity because the two-phase routing is applied independently at each decoding step. In terms of accuracy, performance appears to be largely preserved, with only minor degradation in Qwen3-235B, and in other cases even slight improvements over the vanilla baseline with careful tuning.

The reviews are somewhat mixed. Several concerns regarding the strength of the claim regarding "no statistically significant degradation" as well as the limited evaluation across tasks and sensitivity to batch size has been pointed out by Reviewer hN2v and Reviewer D8pW. There were also some doubts about how well the method generalizes to broader benchmarks such as MMLU (Reviewer D8pW, Reviewer vsam). During the rebuttal period, the authors attempted to address these concerns by adding additional MMLU results across different batch sizes. These results suggest that OEA tends to perform better as the batch size increases, which narrows the gap to the baseline. It is also more flexible in practice as the baseline cannot be used at larger batch sizes under the same setting due to KV cache limitations.

In my point of view, this is a nice paper with well-motivated and practical contributions. Particularly, the ability to recover most of the model performance while achieving significant latency reduction is non-trivial. Therefore, I recommend acceptance and belive that this paper will make a strong contribution to advancing large moel and make these models more appliccable to real world scenerio.